# Epigenetics and Pregnancy: Conditional Snapshot or Rolling Event

**DOI:** 10.3390/ijms232012698

**Published:** 2022-10-21

**Authors:** Mariana Andrawus, Lital Sharvit, Gil Atzmon

**Affiliations:** Department of Human Biology, University of Haifa, Haifa 3498838, Israel

**Keywords:** epigenetics modification, pregnancy, DNA methylation, maternal stress, pre-eclampsia, inheritance

## Abstract

Epigenetics modification such as DNA methylation can affect maternal health during the gestation period. Furthermore, pregnancy can drive a range of physiological and molecular changes that have the potential to contribute to pathological conditions. Pregnancy-related risk factors include multiple environmental, behavioral, and hereditary factors that can impact maternal DNA methylation with long-lasting consequences. Identification of the epigenetic patterns linked to poor pregnancy outcomes is crucial since changes in DNA methylation patterns can have long-term effects. In this review, we provide an overview of the epigenetic changes that influence pregnancy-related molecular programming such as gestational diabetes, immune response, and pre-eclampsia, in an effort to close the gap in current understanding regarding interactions between the environment, the genetics of the fetus, and the pregnant woman.

## 1. Epigenetic Reorganization during Pregnancy and Embryogenesis Processes

### 1.1. Epigenetics

Epigenetic inheritance can be defined as heritable changes in gene expression or cellular phenotypes that occur without altering the underlying DNA sequence [1,2]. In most cases, these epigenetic modifications produce reversible changes in gene function, influencing gene expression through several mechanisms. These mechanisms governing epigenetic changes include DNA methylation, which usually impacts cytosine guanine dinucleotide (CpG) motifs, as well as post-translational modifications (PTM) of the amino-terminal (N-terminal) tails of histones. The most common PTM are methylation, acetylation, phosphorylation, and ubiquitination, while sumoylation, ADP-ribosylation, deimination, and proline isomerization are more recent discoveries that have yet to be thoroughly investigated [3,4]. Other epigenetic regulatory mechanisms include RNA regulation, through the ability to serve as enhancers of transcription, decoy microRNA (miRNA) targets, and mediators of chromatin-modifying complex recruitment to specific locations [5]. Epigenetic modification is assumed to be a consequence of changes in an organism’s environment that cause changes in biological processes that control heritable changes in gene expression. Some environmental alterations that have been linked to epigenetic changes include starvation and various chemical exposures. Moreover, environmental exposures are responsible for causing epigenetic changes that lead to a disease process. DNA methylation is the most studied type of epigenetic modification, and it has been widely assessed in pregnant women [6].

### 1.2. Pregnancy and Epigenetic Changes

During pregnancy, physiological changes occur to nurture the developing fetus and prepare the mother for labor and delivery. A growing body of evidence indicates that maternal lifestyle and prenatal factors are associated with serious health consequences and diseases later in life [7]. Pregnancy is characterized by substantial physiological alterations affecting processes including the immune system and the metabolism of glucose and fats. Epigenetic mechanisms such as DNA methylation, chromatin modifications, and the modulation of gene expression during gestation are believed to contribute to the development of various diseases and disorders. However, far less attention has been paid to the effects of pregnancy on the mother, such as chromatin epigenetic changes that are an important component of long-term responses to hormonal signaling [3].

Tissues and organs during stages of pre-conception, egg fertilization, pregnancy, and the first years of the newborn life are susceptible to environmental and lifestyle factors that protect the organism from susceptibility to disease. Several studies have offered insight regarding the impact of early life stress during the developmental stages on adverse pregnancy outcomes [8,9]. These factors modulate cellular function and gene expression through mechanisms including DNA methylation and histone modification. These epigenetic mechanisms can serve as central regulators of systemic physiological and biological processes, diseases, and placental development and function [10]. Only recently have studies begun to explore the effect on the epigenome during gestation, although epigenetic biomarkers have been previously used to define the pathophysiology of pregnancy complications and adverse early pregnancy outcomes (Figure 1) [11].

#### 1.2.1. DNA Methylation

DNA methylation occurs through the addition of a methyl group at the C-5 position of cytosine in the context of a CpG dinucleotide. Methylated DNA is usually found in areas of the genome that are inactive or silent, whereas unmethylated regions correspond to actively transcribed DNA areas [7]. Various cellular processes, such as genomic imprinting, chromosomal stability, chromatin structure, embryonic development, cellular differentiation, X-chromosome inactivation, and transcription are regulated by DNA methylation [8]. Epigenetic regulation has been shown to be linked to human health, primarily because certain modifications established during early development are labile or metastable and modifiable by environmental factors. CpG methylation is linked with gene silencing, and clusters of CpG sites, known as CGI (CPI), are often non-methylated [9]. Gene expression patterns can thus be modified through the methylation of the DNA encoding specific genes.

The enzymes responsible for DNA methylation include the methyltransferases (DNMTs) DNMT1, DNMT3A, and DNMT3B. DNMT1 duplicates the pattern of DNA methylation during genomic replication and is required for the maintenance of all methylation in the genome [10]. DNMT3A and DNMT3B induce de novo DNA methylation and are responsible for establishing new methylation patterns [11,12,13]. DNMT1 is a maintenance methyltransferase that plays a role in placental development and can disrupt this process in the first trimester of pregnancy [14]. Several studies have suggested that TET protein regulates DNA methylation and gene expression through a variety of mechanisms (TET family proteins and 5-hydroxymethylcytosine in development and disease). The TET proteins have three members, TET1, TET2, and TET3, which are responsible for the conversion of 5-mC to 5-hmC. TET1 and TET3 have the DNA-binding domain CXXC to recognize CpG sites, such as DNMTs, while TET2 lacks this domain. Expression levels of TET1 and TET2 are high in embryonic stem cells and decreased dramatically when differentiation occurs. TET1 and TET2 involve regulating pluripotency in stem cells through a reciprocal feedback process. TET1 and TET2 control promoter hypo-methylation of pluripotency-maintaining genes to maintain the expression on embryonic stem cells [15]. TET2 is an enzyme that has a necessary role in the regulation of trophoblast cell invasion through regulating matrix metalloproteinase-9 (MMP-9) promoter demethylation at the −233 CpG and −712 CpG sites in the promoter region [16]. Folate, betaine, choline, and methionine are considered higher methyl-group donors during pregnancy. Pauwels et al. examined the alteration in methyl-group donor intake and the DNA methylation pattern in the blood cells of pregnant women. They found that until 12 weeks pre-pregnancy, there was a high intake of methionine, choline, and betaine, and high consumption of three methyl-group donors, as well as in the 2nd and 3rd trimesters of pregnancy. Therefore, pregnancy-related methylation profiling and its association with methyl-group absorption in a healthy population has improved the current understanding of the growth of pregnancy-related disorders [17]. Badon et al. studied maternal peripheral blood samples and explored the association between leisure-time physical activity (LTPA) and DNA methylation. They proposed that LTPA may influence maternal epigenetic biomarkers, possibly in an offspring sex-specific manner [10].

Although there is emerging evidence regarding DNA methylation variability over time, little is known about the dynamics of DNA methylation status during pregnancy. Gruzieva et al. performed an epigenome-wide analysis evaluating temporal changes in DNA methylation associated with pregnancy testing and blood cell type correction. Certain key fetal developmental windows exist when dietary factors have a pronounced impact on the pregnancy-related epigenome. For instance, the epigenome is most susceptible to environmental factors during embryogenesis owing to the extremely high rate of DNA synthesis. During this period, DNA methylation patterns are established which are essential for normal development and differentiation [18]. However, it remains unclear to what extent maternal consumption of methyl groups during pregnancy may affect maternal DNA methylation. Furthermore, researchers that assess maternal DNA methylation are generally interested in detecting pregnancy-related illnesses and preterm birth to provide a better understanding of how epigenetic variations during pregnancy can help to explain the biological mechanisms underlying important physiological alterations and adaptations needed for appropriate fetal development and preparation of the mother for childbirth and for the postnatal period [18]. For example, Anderson et al. reported the existence of significant differences in the first trimester DNA methylation patterns of maternal white blood cells collected from pregnancies complicated by pre-eclampsia (PE) compared to normal pregnancies [19]. Moreover, Enquobahrie et al. found that maternal DNA methylation patterns in blood samples collected after 16 weeks of pregnancy were different in women who had two consecutive pregnancies both complicated with gestational diabetes mellitus (GDM) relative to women that had only one pregnancy complicated by GDM [20]. In addition, Burris et al. found that higher early pregnancy maternal LINE-1 element (long interspersed nucleotide element-1, as a transposon) methylation in white blood cells predicts a lower risk of preterm birth [21]. Gruzieva et al. performed an epigenome-wide longitudinal DNA methylation study (EWAS) of a well-characterized sample before, during, and after pregnancy, and identified 196 CpG sites displaying intra-individual longitudinal changes in DNA methylation. Most of the differentially methylated genes exhibited a decrease in the average methylation levels over the studied period. For instance, the Hunger Winter Families study found that the adult mothers who were prenatally exposed to famine harbored hypo-methylation patterns in the maternally imprinted insulin-like growth factor 2 (*IGF-2*) gene, a key factor involved in human growth and development [22,23,24]. One study compared obese pregnant women with normal weight pregnant women and found a decrease in gene expression levels of TET-1, TET-2, and TET-3, while the expression of DNMTs including DNMT1 and DNMT3A and -3B remained unchanged. These results suggest that hypoxia-induced genome-wide DNA hypermethylation is performed by delay DNA demethylation rather than DNMTs-mediated catalyzed methylation in CpG dinucleotides [25]. One of the basic aspects of pregnancy is the complex interplay of epigenetic changes, whose proper combination and critical timing are essential for a successful pregnancy.

Pregnancy-related risk factors can influence maternal DNA methylation patterns, potentially increasing clinical risk and rendering mothers susceptible to long-term consequences. There is an increasing need to investigate the genetic and epigenetic factors such as DNAm from obstructive maternal tissues such as whole blood, which can help monitor fetal–maternal conditions during pregnancy. During pregnancy and postpartum, the epigenetic profiling of peripheral blood DNA from pregnant women may improve insight into the physiological alterations that occur. This can help us to recognize potential biomarkers for pregnancy outcomes. Environmental factors and genetic factors have also been assessed regarding their association with each cell type or tissue. Epigenetic differences are regulated in a tissue- and developmental stage-specific manner. Due to the massive between-tissue differences in DNA methylation patterns, it is a considerable challenge to identify instructive epigenetic signatures in blood or placenta epigenomes and then prove that they cause, at least to some extent, methylation variation in the target tissues [26]. A study that was conducted in maternal white blood cells shows changes in the methylation profile of the LINE-1 elements, which are extremely methylated in normal conditions but are hypo-methylated during stress [27]. Hyper-methylation of LINE-1 elements in early stages of pregnancy can lead to preterm birth [21]. In other words, dynamic DNAm changes modulate the differential expression of the genome in variable maternal tissues such as the uterus and breasts, and respond to environmental factors from the nulligravid to postpartum state [28]. Several environmental and genetic factors are presented in Figure 2.

#### 1.2.2. Pregnancy and the Immune System

Immune tolerance is induced during pregnancy to ensure that mothers do not reject the semi-allogenic fetus. Accordingly, the first and early second trimesters of pregnancy are considered to correspond to an inflammatory phase [29,30]. Leukocytes play a critical role in the proliferation, inflammation, immune tolerance, and maternal adaptation processes that are integral to normal placental and fetal development. White et al. found evidence of global pregnancy-related hypo-methylation in maternal leukocytes relative to that observed in non-pregnant individuals, leading to the identification of candidate genes (Figure 3) that are involved in maternal adaptation to pregnancy based on their altered methylation profiles. These included hypo-methylated immunity-related genes (IL1R2, IL1-β, and HPR), gametogenesis-related genes (SPAG4, CCIN), and housekeeping genes (PC, NDFUS2). The involvement of interleukin (IL)-1 family genes has previously been reported to be important in the establishment and progression of pregnancy. These authors further found that the hypo-methylation of the haptoglobin-related protein (HPR) gene may contribute to increases in HPR expression during normal pregnancy. HPR is a secreted plasma protein associated with high-density lipoprotein (HDL) particles.

#### 1.2.3. Pregnancy and Stress

Many studies have identified changes in DNA methylation as a possible mediator of the effect of prenatal stress on offspring. Babenko et al., for example, hypothesized that prenatal stress has epigenetically regulated effects on health and diseases of the nervous system from early development to old age [31]. For example, in placentas affected by pre-eclampsia, increased DNA methylation has been observed at CpG residues associated with the genes encoding glucocorticoid receptor (GR) and corticotropin-releasing hormone-binding protein (CRH-BP), compared with normal placenta tissues [8,32]. Pregnancy is guided by a well-known interplay of hormonal changes, such as estradiol and estrogen, that not only ensures that key features of the developmental process such as the placenta occur at the right time, but also that birth occurs after the fetal development is complete. Following ligand binding by alpha and beta estrogen receptors from the cell membrane to the nucleus, where they bind directly, promotion of gene expression occurs by the interaction of several epigenetic enzymes. This results in the establishment of multimeric protein complexes targeted to specific genomic locations, which leads to an epigenetic signaling cascade [3,33,34].

#### 1.2.4. Pregnancy and Histone Modification

Histone modification is another epigenetic modification that can affect pregnant women. Histone modifications are considered an essential component of the higher-order structure of chromatin and the impact of the signaling molecules on the maternal body, specifically on diverse organ systems. Histone deacetylation, one of the most typical histone modifications, is linked to gene silencing, whereby histone acetylation is related to transcriptional activation. Histones may undergo numerous categories of modifications, such as acetylation, methylation, or phosphorylation that affect chromatin structure and gene expression. Studies into genetic variability proposes that the level of histone demethylation varied until 20 weeks of postpartum. During this period, demethylation levels of four histones (H3K9, H3K27, H3K36 and H3K79) differed from the levels of non-pregnant women, suggesting that they had not reverted to pre-pregnancy demethylation levels during the 20 weeks of postpartum [12].

Michalczyk et al. also demonstrated that lower overall histone lysine demethylation of H3K27 and H3K4 in women with GDM who subsequently developed T2DM was associated with women with GDM who did not. In addition, variation of histone lysine demethylation levels in pregnant women with GDM compared with the postpartum period was observed [12].

#### 1.2.5. Pregnancy and Non-Coding RNA

The third epigenetic modification involves non-coding RNAs (ncRNA), which also participate in the epigenetic regulation of gene transcription. Non-coding RNAs are divided into two groups based on their sizes: small ncRNAs, including microRNA (miRNA), piwi-interacting RNA, small interfering RNA, and small nucleolar RNA, generally function as negative regulators of gene expression; and long ncRNAs. Some evidence has shown the functional diversity of ncRNAs; in particular, long non-coding RNAs regulated disease and development. Non-coding RNA is highly expressed in the placenta in early pregnancy, particularly in trophoblasts, and can play a major role in placentation. The expression levels of maternal ncRNAs in the placenta in early pregnancy are expected to predict the onset of preterm onset preeclampsia (POPE). Despite the fact that many ncRNAs have been proposed as potential PE predictors, the main focus was devoted to examining their expression in the maternal blood and placenta of post-onset PE and week-adjusted control pregnant women. Maternal–fetal and environmental factors influenced by epigenetic mechanisms in pregnancy are a dynamic phenomenon that can affect and change embryo-fetal development during the phases of gestation [13].

Considerable evidence has gradually shown that the pregnancy environment regulates an extraordinary effect at the epigenetic level, with at least two diverse mechanisms: by direct regulation of the stages of implantation and placentation, and extensive shaping of epigenetic patterns during prenatal progress, decisive long-term consequences occur in the offspring [13,14,15,16] Mammals undergo two-step DNA methylation reprogramming that affects over 80% of the genome through gametogenesis, and then over the initial moments after fertilization [13]. Results have also confirmed the genomic methylation status to be functionally correlated with fetal intolerance. A study on the methylation status of *SLC9B1* has established that the methylation pattern is able to predict the medical disorder of possible pregnancies related to fetal intolerance [35].

### 1.3. Embryo Development

Two epigenetic modifications, DNA methylation and histone modification, play a main role in embryonic and fetal development, including the determination and maintenance of cells, and in the regulation of genomic imprinting [36,37]. During the early stages of embryonic development, the DNA methylation profile goes through reprogramming. Conversely, through the preimplantation phase, epigenetic reprogramming shows differential impacts on the embryo and the primordial phase of gametogenesis. During this time, the embryo endures genome-wide demethylation; firstly active demethylation of epigenetic reprogramming, followed by fertilization and formation of the blastocyst. Passive demethylation occurs because of the lack of maintenance methylation during DNA replication [38,39].

Some evidence has suggested that premature birth and low birthweight is caused by maternal stress during pregnancy [40]. The hypothalamic–pituitary–adrenal (HPA) axis is the main mechanism that secretes steroid hormones, connecting maternal stress during pregnancy to fetal development and birth outcome [41]. Cortisol is considered a mediator in the linking between maternal stress during gestation and child outcomes suggested in some studies. The study of Obel et al. established that women who experienced high levels of pregnancy-specific stress as a result of their fear of pregnancy complications had lower morning cortisol levels at 14 gestational weeks, and higher levels in advanced pregnancy, at 30 gestational weeks. Despite the prior findings linking the association between maternal stress hormone levels and preterm delivery and shorter gestational periods [42], the Smew et al. study demonstrated that maternal stress through gestation was not related with birthweight [17]. This was in accordance to Bolten’s study, which did not find a link between prenatal distress and cortisol level [42,43].

### 1.4. Epigenetic Changes in the Placenta

The placenta, as an essential regulator of fetal growth, survival, and development, is likely to play a key role in controlling fetal nutrient metabolism through epigenetic mechanisms, primarily through maternal genomic imprinting. For example, GDM are associated with changes in placental anatomy and physiology, leading to disruptions in the supply of placental nutrition.

It is becoming increasingly clear that proper epigenetic regulation is a significant mediator of placental development and function, although the specific role of epigenetic factors in the pathogenesis of hypertension-linked placental dysfunction remains incompletely understood. Epigenetic mechanisms support placental functional adaptation by altering environmental conditions. The most robust evidence regarding the impact of epigenetic factors during pregnancy is derived from early studies exploring human embryonic and placental methylomes, revealing that the placenta maintains a globally hypo-methylated DNA profile throughout pregnancy, with such hypo-methylation typically occurring in large domains known as partially methylated domains (PMDs) that cover almost 40% of the placental genome [44]. Epigenetic changes in the placenta may provide an effective mechanism that links environmental stressors to adverse pregnancy outcomes, particularly in the context of fetal malformations. Epigenetic changes have been demonstrated within placental tissues, including altered DNA methylation patterns, changes in the binding activity of DNA-binding proteins (DBPs), imprinted regions in the *H19/IGF2* genes, and shifts in the methylation of histones associated with the human growth hormone gene in placental chromatin [45]. In particular, imprinted genes in mammals are unevenly expressed from the two parental alleles. This is the case of the *H19/IGF2* locus, which consists of the maternally expressed *H19* non-coding RNA and the paternally expressed *IGF2*. The epigenetic regulation of the *H19/IGF2* gene-cluster relies on a shared set of enhancers located downstream of the *H19* gene. On the maternal chromosome, *H19* gene expression is driven by this set of regulatory enhancers, whereas on the paternal chromosome, the enhancers are used by *IGF2* (Figure 4). Further investigations are ongoing to more fully clarify the epigenomic modulation of pregnancy-related pathology, with a particular focus on the epigenetic profiles of placental resident immune cells, which are primarily responsible for the safe onset and progression of pregnancy.

## 2. Epigenetic Changes Characterizing Maternal Pregnancy-Associated Disorders

### 2.1. The Epigenetics of Pre-Eclampsia

Pre-eclampsia (PE) is a complex, heterogeneous disorder that affects 2–10% of pregnancies and is characterized by a combination of hypertension, proteinuria, and edema after 20 weeks of gestation, potentially resulting in maternal multi-organ dysfunction and causing 60,000 maternal deaths worldwide annually [46]. PE is one of the leading causes of maternal mortality. In PE placentas, altered global DNA methylation profiles are negatively correlated with maternal blood pressure [47,48]. Abnormal DNA methylation during pregnancy can contribute to the hypertensive irregularities underlying the pathogenesis of PE [49], with PE-related DNA methylation being particularly important in this context [49,50]. Both Kamrani et al. and Huang et al. conducted research on the epigenetics of PE, using the placenta and cell-free fetal DNA in maternal blood. Kamrani et al. reported that the hyper-methylation of several genes plays a role in the development of PE [29]. For example, under PE conditions, *IGF-1* was considerably decreased in hypoxic trophoblasts, and pyrophosphate sequencing showed that the *IGF-1* promoter was hypermethylated in PE placentas mediated by DNMT-1 [49]. *VHL* is a tumor suppressor protein that is downregulated in PE, and this protein is critical for proper placental development. Moreover, the study of Huang et al. showed that PE linked with raised placental intracellular adenosine-mediated DNA hypermethylation altered placental gene expression [51]. Studies have shown that hyper-methylation of tsyncytin-1 decreased expression levels in PE, caused intrauterine growth restriction, and elevated liver and low platelets syndrome [52,53]. Generally, dysregulation of various genes’ methylation can give rise to PE for several reasons, for instance, hypoxia, oxygen homeostasis in early PE, hypoxic trophoblasts, and infiltrating neutrophils from maternal systemic blood vessels. Separately, Anderson et al. investigated maternal peripheral white blood cells and placental chorionic tissue samples from normotensive women and individuals with PE [54]. Genome-wide DNA methylation analyses revealed that 64% and 36% of differentially methylated sites were associated with major increases and reductions in methylation, respectively. Pathway analyses suggested that differentially methylated genes were associated with cell signal transduction processes pertaining to lipid binding, protease enzyme inhibition, protein–protein interaction, cell cycle processes, and adhesion. Associations with signaling pathways including cellular metabolic processes were predominant for genes exhibiting significant reductions in methylation [19]. For a further summary of PE-related changes in DNA methylation, see Table 1. Several studies have established the role of methylation in normal and PE pregnancies by revealing that maternal leukocyte DNA methylation is associated with maternal adaptations critical for normal pregnancy outcomes, explaining why genome-wide methylation profiles of maternal leukocyte DNA at the time of delivery exhibit increased methylation in women affected by PE as compared to normotensive pregnant women, and underscoring the potential role of such differential methylation in the regulation of PE-related changes in gene expression. Recent work also supports a role of PE-associated epigenetic changes by demonstrating a link between PE and the acceleration of epigenetic aging [55,56,57].

### 2.2. DNA Methylation in Placental Pathology and Senescence

#### 2.2.1. Mitochondrial Changes during Pregnancy

Early placental aging is linked with aberrant changes in telomere length, cellular senescence, and mitochondrial dysfunction [63]. Recent evidence supports that mitochondria play important roles in physiological adaptations during gestation, and mitochondrial functional variations have been reported among healthy and complicated gestations. Their dynamic element is critical in regulated central cellular activities, and becomes the major source of maternal metabolic energy [64]. Mitochondria are essential organelles for various aspects of cellular homeostasis, as well as cellular energy production, regulation of apoptosis, and maintenance of calcium homeostasis. Some studies have observed differences in mitochondrial adaptation when comparing mitochondria associated with healthy pregnancies to those of non-pregnant women [65,66]. Mammalian mtDNA is a rounded double-stranded molecule, and determines only 37 genes, 13 of which are respiratory chain subunits, while 24 are RNA components, such as tRNAs and rRNAs. The mitochondria provide the cell energy and are involved in multiple cellular processes, including oxidative phosphorylation and ATP production, b-fatty acid oxidation, the citric acid cycle, and the regulation of apoptosis. Mitochondrial epigenetics is an evolving area that explores how mtDNA methylation may impact development, health, and disease later in life. It should also be postulated that, as mitochondria is a major sole maternal inheritance (despite some evidence for mitochondrial sperm inheritance), the study of mtDNA methylation inheritance and its impacts on disease risk occurrence could be of significant benefit (mitochondrial epigenetics and environmental exposure). Some publications contribute to the shift in this opinion by supplying available technical developments, in addition to information on regional CpG and non-CpG mtDNA methylation, regulation by methyltransferases, and relations with diseases [67,68,69].

The epigenetic regulation of mitochondrial function is an area of active study, and three key mechanisms have been reported to control gene expression within mitochondria, including DNA methylation, non-coding RNAs, and post-translational changes in nucleoid-related proteins. Epigenetic and post-translational modifications facilitate and regulate mtDNA and mitochondrial protein cross-talk among the nucleus and mitochondria, and eventually give rise to the maintenance of cell health and homeostasis [70].

Mitochondria methylation within CpG dinucleotides as CGI are virtually absent due to the smaller size of mitochondrial DNA (mtDNA) and its short non-coding control region. Non-coding RNAs levels can be manipulate cellular and mitochondrial function during diverse forms of cancer, making them a potential target for cancer therapy [71,72]. Post-translational changes include lysine methylation, phosphorylation, acetylation, ubiquitylation, sumoylation, PARylation, and a number of kinases and protein phosphatases have also been identified within the mitochondria [44].

Mitochondrial DNA levels in the maternal peripheral blood of women affected by PE were initially reported to be associated with high levels of oxidative stress and mitochondrial dysfunction [45]. More recent work exploring the existence and biological relevance of mtDNA methylation profiles has yielded inconsistent findings. Several novel studies have investigated how airborne particulate matter such as cigarette smoke can influence the methylation of gDNA and/or mtDNA [73]. Overall, it is likely that epigenetic changes in specific targeted locations on the mitochondrial chromosome may affect transcription and/or replication of mtDNA, subsequent mitochondrial gene expression, and associated oxidative phosphorylation. These changes may impact the appropriate functioning of placental trophoblasts, and may thus play distinct roles in the contexts of placental activity and fetal development. Mitochondria include the machinery needed to epigenetically alter mtDNA and the precise mechanisms liable for mitochondrial methylation alteration, and eventually impact on fetal growth and remain decisive. However, areas of potential interest for examination can include intrauterine hypoxia or certain chemical components [74]. Accordingly, it is hypothesized that transcription and/or replication of the mitochondrial DNA following mitochondrial gene expression, and possibly even control of oxidative phosphorylation, are influenced by epigenetic modifications at specific targeted locations on the mitochondrial chromosome. These might affect the proper functioning of placental trophoblasts, and as a result could have several effects on placental activity and fetal growth.

Even though the fact that earlier findings could have overestimated mtDNA methylation [75], the study of Shock et al. suggests that substantial levels of methylation can occur on specific sites contained by the mtDNA. This is supported by exploration regarding the activity of mitochondria-localized DNA methyltransferase 1 (mtDNMT1) enzymes, which have been shown to bind mtDNA and may supply the mechanism by which mtDNA is methylated [76]. Recent study has shown that mtDNA is widely methylated at non-CpG sites [77,78,79], which might cause pyrosequencing-related PCR biases [67].

A main factor in mitochondrial adaptation in the placenta is the oxygen saturation of the placental tissue, which changes continuously during pregnancy. Some variations in oxygen tension enable key signaling events that drive developmental stages; however, those alterations in oxygen tension contribute to the production of ROS, which in surplus can be damaging (Figure 5). A change in oxygen concentration has been shown to regulate aging phenotypes in the Swan-71 placental trophoblast cell line (the Swan 71 cells express HLA-G and secrete low levels of human chorionic gonadotropin (hCG)). If variations in oxygen concentration occurs suddenly, it can lead to a change in cell signaling and amplify damage, indicating that the placenta is no longer senescent chronologically and causing placental pathologies [80].

The maintenance of DNA methylation at sites not related to known imprinted genes appears to be important for trophoblast development. Human placental DNA methylation levels rise considerably with prenatal age, possibly reflecting a gradual loss of cellular plasticity often related to higher DNA methylation levels as cells terminally differentiate. When cells finally differentiate, a special feature of the trophoblast epigenome is the frequency of large areas, so-called intermediate methylation, which extend across transcriptionally repressed, polycomb-focused regions of the genome. This function of the epigenome is still unknown [81,82,83,84].

#### 2.2.2. The Relationship between Cellular Senescence and Adverse Pregnancy Outcomes

Senescence can occur in response to a range of physiological stressors and associated molecular damage [85]. These stress signals include oxidative stress, which is an important contributing factor underlying the pathophysiology of complex pregnancies. There is evidence that pregnancies affected by PE may be distinct from normotensive pregnancies with respect to epigenetic markers of aging and senescence in tissues and organs, with PE and pregnancy-related outcomes including gestational length and birth weight being linked to accelerated epigenetic aging during pregnancy in women with PE, as established by the application of the “epigenetic clock”, whereas no comparable changes are evident for normotensive pregnancies [86]. In pregnancies affected by PE, the trophoblastic plugs open prematurely, contributing to an influx of arterial blood that increases stress in the placenta and can lead to defective placentation.

There have been a limited number of studies to date suggesting that prenatal markers of maternal biological age such as telomere length, an indicator of cellular senescence, may be linked with pregnancy outcomes. Many studies have explored the effects of maternal dyslipidemia on the epigenetic aging of the placenta [87,88,89]. Placental aging is a complicated process that depends on factors including genetics, epigenetics, inflammation and related diseases, sex hormone levels, oxidative stress, and external environmental factors such as diet, exercise, and socio-demographic parameters. While optimal placental aging corresponds with gestational age and complements optimal fetal growth and development, premature placental aging is correlated with unfavorable obstetric complications including early-onset PE, low birth weight, stillbirth, and preterm birth. As such, efforts aimed at controlling dyslipidemia in early pregnancy may support the normal aging of the placenta and healthy pregnancy. Cell senescence occurs in the course of the life cycle of all living organisms, including the placenta. It is created, grows, performs its various functions (e.g., endocrine regulation and nourishment of the fetus), and ages. Cellular senescence effects from progressive DNA damage handle the development of the cell’s aging phenotype, and are determined by intrinsic or extrinsic effects. Cell aging may cause apoptosis, developed in the aging of tissues first and then of organs [90]. The resulting age approximated by this clock is described as the epigenetic age, or, more accurately, the DNAm age. The contrast between epigenetic age and chronological age, which reflects the rate of epigenetic aging, has biological importance; enhanced epigenetic aging is linked to several age-related disorders and conditions. The Giller et al. study defined that pregnancy is a model for aging, and induces age-related diseases, for instance, diabetes and heart disease, and other similar mechanisms that are associated with aging [91].

As shown by published findings showing abnormal senescence and heightened senescence in PE placentas, an enhanced senescence-like state can play a mechanistic role in the pathophysiology of PE. It showed that, once recognized, senescence may spread from cell to cell, indicating that placental aging may cause senescence of distant maternal organs and cells. The study of Poganik et al. demonstrated that both human and mice pregnancy stimulates a reversible increase in biological age, peaking across delivery and resolving post-partum [92]. Oxidative stress stimulates the activation of repair pathways, inhibits cellular proliferation through the induction of senescence or transient cell cycle arrest, and can drive apoptotic cell death [93]. Oxidative stress is characterized as an imbalance between the generation of reactive oxygen species (ROS) and the ability of antioxidant defense mechanisms to mitigate ROS-induced damage. Oxidative stress is recognized as a key factor in the pathogenesis of adverse pregnancy outcomes. The accumulation of oxidative stress causes damage to lipids, proteins, and DNA in the placental tissue, which induces a form of accelerated ageing. Many studies indicated that oxidative stress and placental ageing can play a role in the pathophysiology of complicated pregnancies, with a specific importance on pregnancy complicated by spontaneous preterm birth, intrauterine growth restriction, pre-eclampsia, pregnancy loss and stillbirth [94]. Even though a physiological balance between ROS and antioxidant activity is maintained in normal pregnancies, an imbalance may increase oxidative stress. The placenta experiences an increased level of oxidative stress in certain pathological pregnancies, particularly those complicated by maternal smoking, gestational diabetes, fetal growth restriction, preeclampsia, and miscarriage.

### 2.3. Epigenetic Modifications Associated with Obesity and Gestational Diabetes

GDM is the most common metabolic condition, and may cause complication for the mother in the short- and long-term during gestation. GDM can be defined as diabetes diagnosed during the second or third trimester of pregnancy in women unaffected by overt diabetes prior to gestation, with the etiology of GDM being similar to that of type 2 diabetes mellitus (T2DM) [95]. Epigenetic mechanisms can increase the risk of becoming obese and diabetic, and can be influenced by the nutritional status and physical activity patterns of the parent. Many environmental abuses such as restriction or unhealthy nutrition, lack of exercise, tobacco smoking, alcohol consumption, exposure to environmental pollutants, psychological stress, ethnicity, and hypertension have been recognized to increase an individual’s risk of metabolic disorder during their lifetime in around 50% of cases [95,96,97,98]. The prevalence of GDM has risen dramatically by over 30% within the last two decades in several countries, including developing countries [99]. One of the main reasons underlying this increased incidence can be attributed to advanced pregnancy, which in turn is associated with the presence of risk factors in pregnant women, including being overweight or obese, that render them more susceptible to experiencing hyperglycemia during pregnancy. As some non-obese women also develop GDM, other factors including unhealthy nutrition and low physical activity before or during pregnancy may also contribute to the risk of this metabolic disease [100,101]. Most studies of DNA methylation in GDM have been conducted using samples of fetal tissue such as the cord blood and placenta. Notably, placental DNA methylation has been found to be correlated with maternal glycemic levels during pregnancy, with increasing maternal glycaemia leading to the demethylation of the leptin gene in the fetus, resulting in increased transcriptional activity and higher leptin levels that may contribute to leptin resistance and the development of obesity [102]. Many previous studies have focused on the epigenetic effects of a single gene on GDM risk, whereas a limited subset of studies have employed a genome-wide approach to assessing the different methylation profiles of the placenta and cord blood in GDM and non-GDM participant groups [103,104]. Kang et al., for example, compared patterns of global methylation between pregnant women and their children in the context of GDM to those of healthy maternal–offspring pairs using blood samples taken at admission, before delivery, and umbilical cord blood samples obtained shortly after delivery [103]. They identified 151 differentially methylated genes in mothers affected by GDM, including the *SLC22A4*, *ADRA1A*, *CACNB2*, and *SERPINE1* genes associated with lipid metabolism pathways. Table 2 presented changes in DNA methylation in placenta function. Changes in the DNA methylation of genes associated with GDM-related metabolic pathways including the JAK (Janus-activated kinase) and MAPK pathways were also noted, with these pathways traditionally being associated with innate immunity, inflammation, and environmental stress responses [103]. Wu et al. further profiled genome-wide DNA methylation changes in maternal peripheral whole blood samples collected from pregnant women 12–16 weeks into pregnancy, prior to the diagnosis of GDM. Through these analyses, the authors found that hook microtubule-tethering protein 2 (*HOOK2*) and retinol dehydrogenase 12 (*RDH1*) which have been mentioned in other studies examining cord blood samples and placenta, were differentially methylated. *HOOK2* codes for a linker protein that mediates binding to organelles and is responsible for the morphogenesis of cilia and endocytosis. *RDH12* encodes a retinal reductase, which also plays a role in the metabolism of short-chain aldehydes. Five additional CpG loci were also identified that were associated with the *COPS8*, *PIK3R5*, *HAAO*, *CCDC124*, and *C5orf34* genes [11].

### 2.4. Epigenetic Modifications and Prenatal Maternal Depression

Depressive disorders are among the leading causes of incapacity worldwide. Depressive indicators during gestation are increased, and around 10% of pregnant women suffer from main depressive disorder (MDD) [107]. These frequencies may be as high as 30% when considering all depressive disorders, and a large proportion of pregnant women experience at least one major depressive event before delivery, with this phenomenon being referred to as prenatal depression. Prenatal maternal depression can cause covalent epigenetic changes in the DNA of their progeny that are discernible at birth in leukocytes and that may be introduced into other tissues, suggesting a model wherein systemic epigenetic changes may be involved in lifelong responses to the in utero psychosocial environment [94]. During pregnancy, maternal depression is associated to a higher risk of obstetric complication such as GDM, hypertension, and PE, as well as smoking, increased alcohol consumption, and inadequate nutrition. Evidence indicates that maternal stress during gestation causes hyper-methylation of *HSD11B2* in the placenta [107].

Bagot et al. found that mothers suffering from depression during the third trimester of pregnancy exhibited hyper-methylation of the glucocorticoid receptor promoter (*NR3C1*) and exon 1f. Interestingly, these effects were not overturned with antidepressant treatment. In addition, prenatal maternal depression changed DNA methylation at the serotonin transporter. Thus, prenatal stress can alter adult susceptibility to depression, partly through alteration in DNA methylation [108,109]. These changes affect specific genes and brain regions, emphasizing the hardness in using peripheral tissues to predict functionally relevant variations within the brain [31]. The study of Guintivano et al. demonstrated that mediated epigenetic change may predict postpartum depression (PPD) risk. PPD occurs during pregnancy or within 4 weeks of delivery, and follows a dramatic decline in the circulating levels of estradiol and progesterone. Interestingly, DNA methylation has been linked to estradiol and can affect mood. The CpG methylation levels at two loci surrounded by *HP1BP3* and *TTC9B* genes were detected as biomarkers to predict PPD status independent of pre-partum mood status. Based on this study, there was a correlation between E2-produced DNA methylation and PPD risk in syntonic regions in humans [110].

### 2.5. Nutrition as an Epigenetic Stimulus during Pregnancy

Nutrition is a key factor that supports a normal pregnancy and consists of both nutrient intake and the prenatal, perinatal, and postnatal composition of the maternal diet. Accordingly, nutrition is one of the most studied and best understood environmental factors shaping epigenetic outcomes.

Nutrients can play a role by inhibiting epigenetic enzymes such as DNMT, histone deacetylases (HDAC), or histone acetyltransferases (HAT), or by changing the accessibility of the substrate required for these enzymatic reactions [100]. HATs catalyze the transfer of an acetyl group from acetyl coenzyme A, while HDACs perform the antagonistic action of removing the acetyl group. This, in turn, changes the expression of essential genes that impacts global health and longevity [111,112]. Mitsuya et al. studied patterns of placental DNA methylation and hydroxy methylation on a genomic scale, and observed a partial but important intersection among the genes that exhibited a rise in DNA methylation and a reciprocal reduction in DNA hydroxy methylation with increased maternal obesity, suggesting a possible reduction in the conversion efficiency of methylation to hydroxy methylation, which is controlled by TET dioxygenases [113]. Other studies have also provided evidence in support of a connection between metabolism and epigenetics. The work by Reichetzeder et al. suggests that the dysregulation of adenosine monophosphate-activated protein kinase (AMPK) and the mammalian target of rapamycin (mTOR) homeostasis are linked to gestational obesity [114,115]. AMPK activation occurs in response to dropping energy levels suggested by high AMP levels and low ATP concentrations. Folic acid is recognized as a one carbon donor for DNA methylation and synthesis. In vivo, the enhanced intake of folic acid-rich foods is associated with the reduced promoter methylation of genes associated with tumor suppression [116]. Additionally, folic acid is required to produce S-adenosyl methionine (SAM), which serves as a methyl group donor in the context of DNA methyltransferase-mediated DNA modification. It has been established that SAM is generated in the cytoplasm and drives epigenetic modifications [117]. Folic acid supplementation is related to global DNA methylation (hydroxyl), in which several studies have shown that there is an inhibition in DNA methylation in response to folic acid consumption. In vivo, the delay in the DNA methylation response to folic acid intake can be partially associated with the extremely slow revenue of folate stores all over the body. The study of Pauwels et al. found that there may be a delay in the DNA methylation response in women who took a folic acid supplement before and during gestation. This would have occurred before gestation [17]. Vitamin D is another micronutrient that alters epigenetic pathways. Low vitamin D concentrations can cause increased inflammation and anti-inflammatory effects due to changes in DNA methylation and histone modifications [118]. Furthermore, severe vitamin D deficiency has been linked with changes in methylation in peripheral blood leukocyte DNA in humans [110]. There are several central windows of fetal development where dietary factors can affect the epigenome during gestation. For example, during embryogenesis, the epigenome is more vulnerable to environmental stressors due to the high DNA synthetic rate, and the DNA methylation patterns required for normal development and differentiation are recognized [119]. However, it is not clear to what extent the maternal intake of methyl-groups through gestation can influence maternal DNA methylation.

### 2.6. Significance of Maternal Immune Responses during Pregnancy

#### 2.6.1. Pregnancy-Related Maternal Immunological Adaptation

The perception that pregnancy is associated with the suppression of the immune system has created a myth of pregnancy as a condition of immunological fragility and hence of enhanced sensitivity to infectious disorder. The maternal immune system undergoes profound transformations during the early stages of gestation [120]. Given their high turnover rates, most immune cells are highly susceptible to environmental alteration and can adapt to a range of stressors. Myeloid-derived cells, in particular, can activate stress response pathways that cause plastic changes in their transcriptional activity. During normal pregnancies, the human decidua contains a considerable number of immune cells, including macrophages, natural killer (NK) cells, and regulatory T cells. Accordingly, immune cells infiltrate the decidua during the first trimester and accumulate around the invading trophoblast cells. As such, pregnancy is more appropriately considered a unique immunomodulatory state, rather than a state of true immunosuppression [121].

Pregnancy hormones are of key importance to the maintenance of pregnancies, and they can profoundly impact the associated immune response. The immunological adaptation in pregnancy is shown in Figure 6. A growing body of evidence has established that sex hormones can regulate most important epigenetic alterations involving the modulation of miRNA expression, DNA methylation status, and chromatin rearrangement. An overview of the current evidence regarding the effects of sex hormones on immune system cells is detailed in Table 3.

#### 2.6.2. Autophagy

Autophagy is a process that maintains cellular homeostasis by eliminating senescent or damaged intracellular organelles and proteins, playing a key role in various pathophysiological processes. The role of autophagy in pregnancy, however, has only been studied to a limited degree. It has been described that basal levels of autophagy contribute to the maintenance of intracellular homeostasis and are necessary for cellular remodeling. Autophagy plays a pivotal role in the processes of embryogenesis, implantation, and maintaining a pregnancy. Emerging evidence suggests that reciprocal interactions exist between autophagy and pregnancy complications. The production of autophagy in particular immune cells results in the stimulation of phagocytic activity in macrophages and the activation of T and B lymphocytes. The thymus is an important organ in which pre-T cells differentiate into mature T cells following positive and negative selection, and autophagy in the thymus has also been shown to shape the T cell repertoire.

Singh et al. were the first to establish diethylstilbestrol (DES)-induced autophagy in the thymus and to highlight the possible role of epigenetic pathways in the process of autophagy. DES is a nonsteroidal estrogen that is categorized as an endocrine disruptor, and these researchers found that DES-induced thymic atrophy was related to increased autophagy in thymocytes and the pronounced upregulation of Becn1 and LC3. DES was additionally found to promote the downregulation of miR-30a expression and to trigger Becn1 hypo-methylation, thereby triggering increased Becn1 expression and inducing autophagy in thymic cells [134]. Elevated autophagic activity in the placenta during pregnancy has previously been shown to be involved in the pathophysiology of PE, with evidence of an inverse association between glucose and placental autophagy [135,136,137]. Finer et al. suggested that placental methylated variable positions (MVPs) were most significantly altered in association with pathways related to placental function and growth, for example, endocytosis and mitogen-activated protein kinase (MAPK) signaling [138]. MAPK signaling is also closely related to autophagic induction, and the underlying signaling mechanisms are commonly disrupted in the context of GDM. The differential methylation and expression of these pathways may thus be indicative of numerous physiological changes occurring in the hyperglycemic placenta, which is stable within a multifactorial disease model [135,137,139,140]. Although the molecular pathways governing autophagy have been well characterized, additional work is needed to fully clarify the regulatory role of epigenetic pathways as regulators of pregnancy-associated autophagic activity.

## 3. Epigenetic Changes during Pregnancy That Affect Offspring Health Later in Life

### The Effects of Epigenetic Inheritance on Pregnancy

The role of developmental and parental environmental exposures in shaping the ultimate metabolic characteristics of offspring has been effectively demonstrated in several human studies of populations exposed to extreme nutrient deficiencies during pregnancy [141]. One of the best-understood instances of epigenetic inheritance pertains to the effects of maternal nurturing behaviors during the first week of life. In fact, the developmental rationale for epigenetic inheritance may be to support a way of adapting quickly to short-term environmental factors without the need to wait for the modification of the underlying genes through mutation and selection.

While epigenetic inheritance is still a rapidly growing field, it remains largely independent of the fundamental models of genetics that are traditionally based on the hereditary role of DNA [142]. Recent human research suggests that diseases resulting from disruptions of the normal epigenetic state (also known as “epimutations”) can be inherited across generations [142]. The Dutch Hunger Winter Families Study, first published in 1976, offers a robust, albeit tragic, human experiment regarding the effects of food restriction in utero on metabolism, cardiovascular health, and age-associated congenital abnormalities. The height of the famine was from the end of 1944 until first quarter of 1945. The approved daily rations varied between 400 and 800 calories per day at the end of 1944 and in the first month of 1945. This study highlighted the significance of the timing of stress exposure on the risk of subsequent disease. Individuals born during this period of famine were at an enhanced risk of schizophrenia and depression, had a more atherogenic plasma lipid profile, responded more to stress, and had a higher risk of coronary heart disease [142]. These individuals also performed worse on cognitive tasks, which may be a sign of accelerated aging. People exposed to famine conditions throughout the entirety of the gestational period suffered from higher rates of type 2 diabetes.

Individuals exposed to hunger only through advanced gestation were born small and continued to be small during their lives, with lower obesity rates as adults compared to those born before and after periods of famine [143]. As such, the impacts of famine seem to be very dependent on its timing during gestation, and the organs and tissues undergoing critical periods of development at that time. However, as noted above, those exposed in early pregnancy suffered high rates of obesity, improved lipid profiles, and changes in cardiovascular disease incidence. Conversely, markers of decreased renal function were particular to those exposed to famine conditions during mid-pregnancy [144]. The Dutch Hunger Winter Families Study provided the first direct evidence for epigenetic programming as a result of prenatal famine exposure. Individuals who had been exposed to the famine periconceptionally were found to exhibit lowered methylation of the *IGF2* gene compared with their unexposed same-sex siblings [23]. This suggests that transient environmental conditions early during human pregnancy can be recognized as persistent alterations in epigenetic evidence that result in lifelong phenotypic impact. Certainly, as previously mentioned, early embryonic development is of particular interest in this regard since it is a crucial period for the construction and maintenance of epigenetic marks.

Feeding mice a diet rich in methyl donors can influence their coat color, body weight, and the health of their offspring [54]. This indicates that transient environmental conditions present early in human gestation can result in the establishment of constant epigenetic changes. Certain inherited metabolic disorders can cause significant maternal complications during pregnancy, and modern epigenetic research raises complex questions regarding maternal responsibility for health [145]. Alterations to the environment during early development can cause permanent modifications in the pattern of epigenetic modifications, as shown in Figure 7.

## 4. Conclusions and Future Perspectives

Pregnancy is a dynamic state, with diverse mechanisms being engaged throughout different trimesters to enable and ensure the effective establishment, maintenance, and appropriate termination of the pregnancy. Pregnancy offers an exceptional platform for examining stress and stress responses, both in the event of pathological and normal pregnancies. Our recent understanding is that epigenetic regulation is a key feature of gestation and development, but it occurs rather slowly. Knowledge of such epigenetic mechanisms may be useful in identifying novel biomarkers for pregnancy-related exposure, burden, or disease risk. Such biomarkers may prove essential for the development of tools for the early detection of risk factors and exposure levels. Here, we have provided an overview of prior studies exploring alterations in maternal DNA methylation throughout pregnancy and related unfavorable conditions. Exploration of epigenomic modification can enable an expanded understanding of the dynamic biological processes that take place during pregnancy. Clarifying how epigenetic regulation depends on early exposure may allow clinicians to identify women at increased risk of adverse pregnancy outcomes and develop precise, personalized, risk-specific interventions. Examination of epigenomic variations can provide an enhanced understanding of the dynamic biological developments occurring during pregnancy. This will enable physicians to classify women at amplified risk of adverse pregnancy outcomes and to develop precise, customized interventions for risk. During prenatal treatment, there may be non-invasive biological markers for early detection of systemic pathophysiological processes (nonspecific tissue) or local processes (specific tissue, such as placenta). Previous investigation has identified epigenetic biological markers specific to the placenta in maternal blood, suggesting that epigenetic biological markers from the placenta may be released or leaked into the mother’s bloodstream during pregnancy. Studies established on whole blood DNAm are generally limited. Whole blood is composed of multiple cell types, and each cell type contributes to the CpG locus-specific DNA methylation signal [96]. Studies on DNAm in whole blood have not topically addressed the issue of cellular heterogeneity. Although there are different methods for adjusting for cellular heterogeneity, they generally use reference methylation data based on an estimate of sample cell proportions [145]. DNA-based deconvolution of selected cell type-specific differentially methylated regions (DMRs) from sorted and purified cell populations of individual subjects has yet to be investigated. Furthermore, during gestation, maternal blood also includes fetal genetic materials, so there is still a necessity to account for both to absolutely parse the maternal component. Another limitation of pregnancy researchers is the sample size; these limitations mean the findings cannot be generalized. Gene-specific or genome wide DNA methylation analyses may supply more insight into mechanisms implicit in the processes prompting preterm birth; while such technologies exist, they continue to be costly for cohort studies with large sample sizes [146]. Lastly, the comparatively high socioeconomic status of the cohort may limit the generalizability of the outcomes [21].

Additionally, while identifying the risk of GDM early in pregnancy (before the onset of fetal overgrowth) is absent, it may be useful in identifying women who may benefit more from intensive prenatal monitoring and interferences (lifestyle or medication) to optimize fetal development and minimize hyperglycemia. The complex evidence published indicates that this emerging field of research should be thoroughly investigated. However, larger and more powerful studies are needed to strengthen the current knowledge.

## Figures and Tables

**Figure 1 ijms-23-12698-f001:**
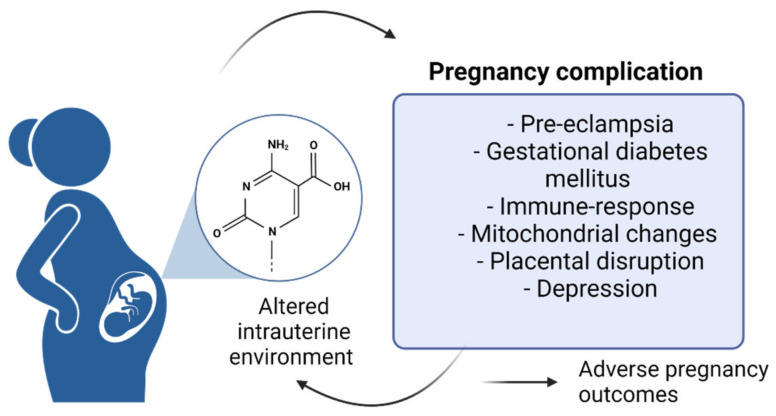
Complications during pregnancy.

**Figure 2 ijms-23-12698-f002:**
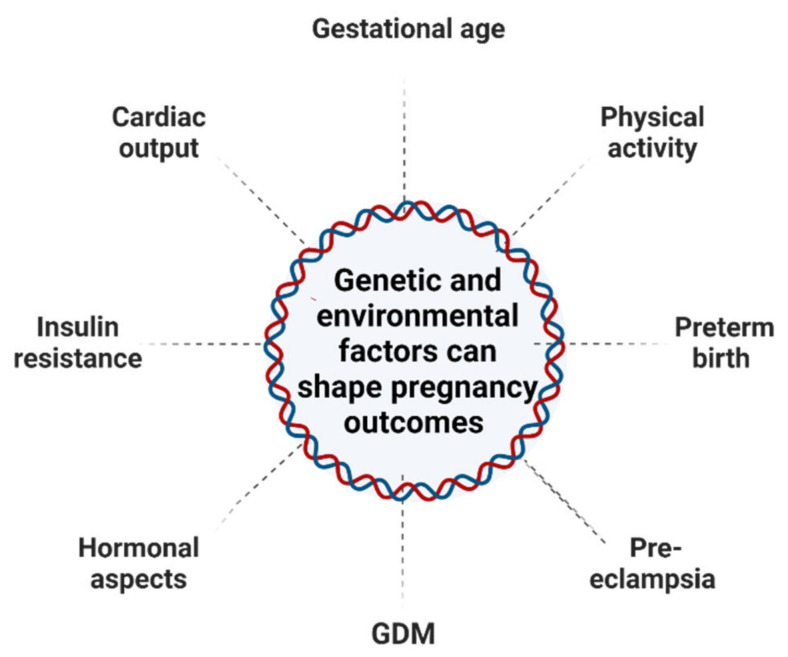
A representative integrative model of the prospective impact of environmental stressors during pregnancy. Maternal lifestyle and exposure to environmental factors; stress, diet, obesity, and physiological changes may modify the epigenome and modulate the epigenome transgenerationally.

**Figure 3 ijms-23-12698-f003:**
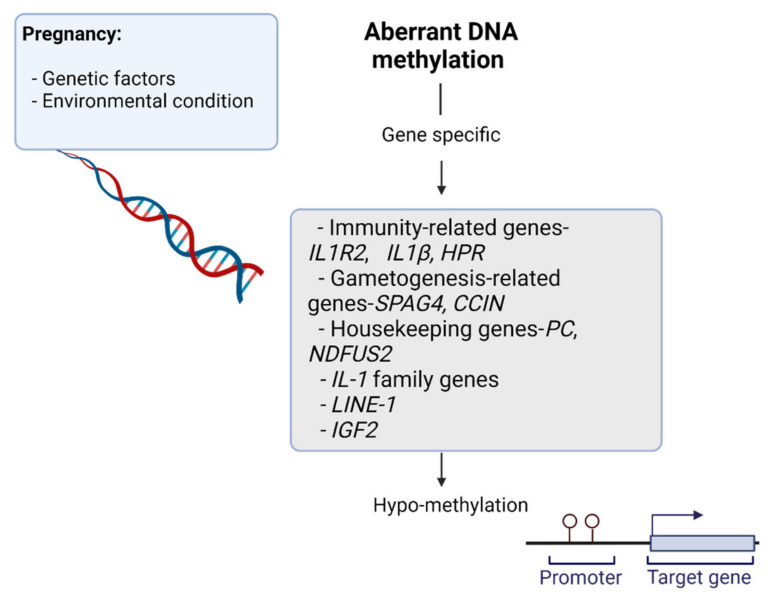
Environmental and genetic risk factors promote changes in DNA methylation. Candidate genes that are involved in maternal adaptation to pregnancy based on their altered methylation profiles may be affected by the epigenetic reprograming process.

**Figure 4 ijms-23-12698-f004:**
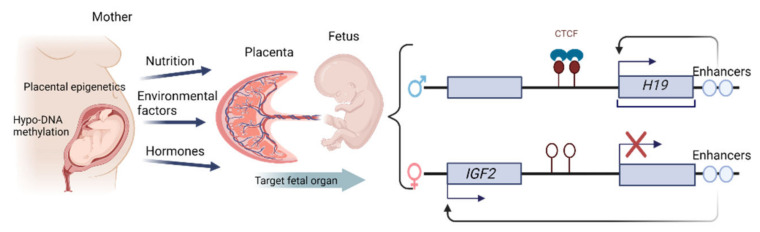
Epigenetic changes have been demonstrated within placental tissues and imprinted regions in the *H19/IGF2* genes. Diagrams illustrate the imprinted genes *H19/IGF2* locus. The enhancer competition model is used to clarify the imprinted rule of *IGF2* and *H19*; the epigenetic regulation of the H19/IGF2 gene-cluster relies on a shared set of enhancers situated behind the H19 gene. On the maternal chromosome, *H19* gene expression is driven by this set of regulatory enhancers, whereas on the paternal chromosome, the enhancers are used by *IGF2*. Large boxes describe genes on the paternal and maternal chromosome, with arrows indicating gene expression. CH3-lollipops represent regions containing methylated cytosine remains. Small boxes display the site of downstream enhancers.

**Figure 5 ijms-23-12698-f005:**
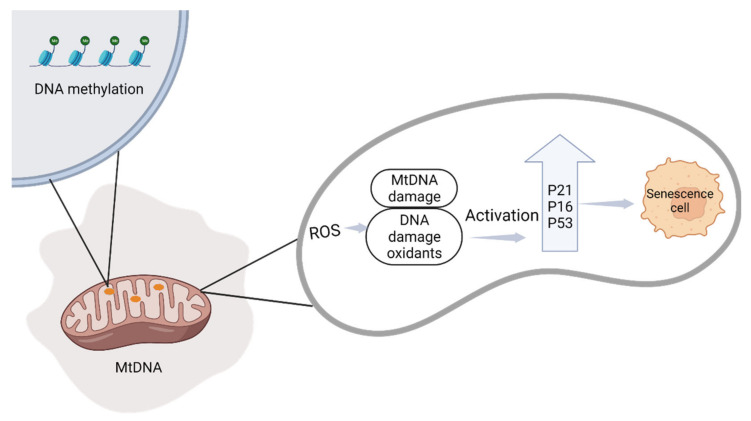
Mitochondrial dysfunction caused by DNA methylation is a potential trigger for mtDNA depletion. ROS production induces translocation of the p53, p16, and p21 proteins to the mitochondria, and activates senescence markers.

**Figure 6 ijms-23-12698-f006:**
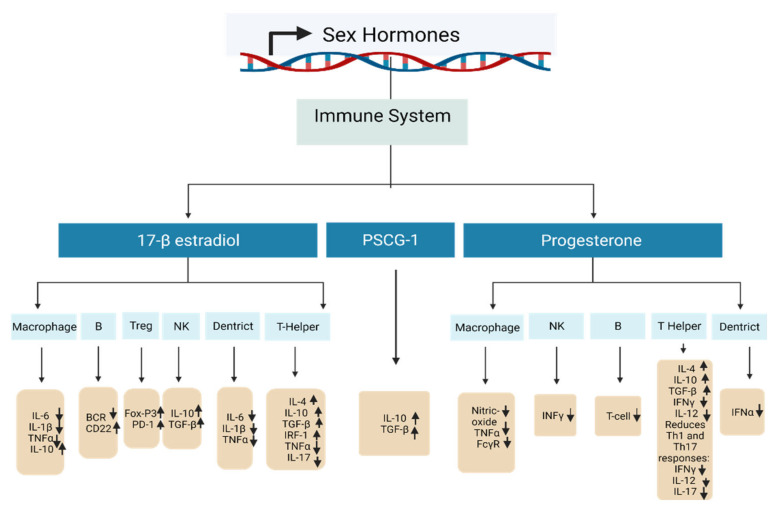
A diagram showing the effects of sex hormones on immune cells in pregnancy. All the major sex hormones in pregnancy support either the proliferation, differentiation, or immunosuppressive function of regulatory T cells (Treg).

**Figure 7 ijms-23-12698-f007:**
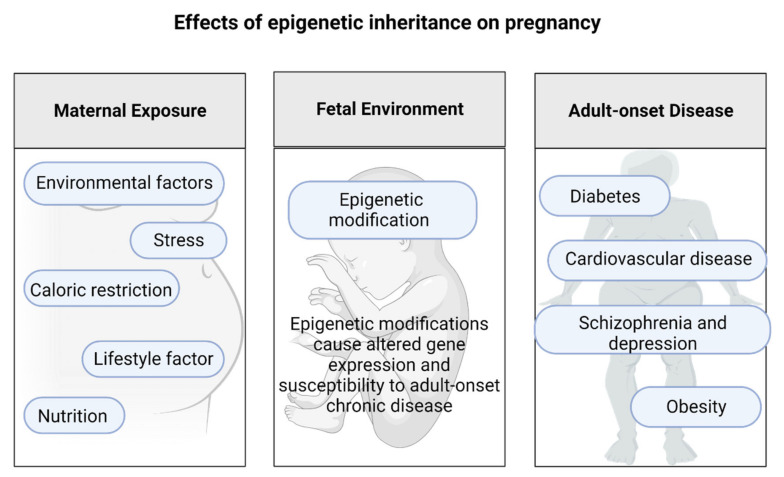
The association between an adverse in utero environment and epigenetic modifications that take place in the developing fetus and may have lifelong consequences.

**Table 1 ijms-23-12698-t001:** PE-related changes in DNA methylation.

Target Genes	DNA Methylation Changes	Functional Roles	Pregnancy Effects	References
*11β-HSD2*	Hypo-methylation	Plays a critical role in hypertension, plays an important role in the regulation of blood pressure by preventing the activation of the mineralocorticoid receptor in tissues such as the placenta.	Pregnancy and maternal distress affect placental *HSD11B4*; during the third trimester, the impact of depression and anxiety symptoms is greater than in the first trimester.	[58,59]
*RUNX3*	Hypo-methylation	Regularly deleted or transcriptionally silenced in cancer; was selected for analysis because tumor progression and gestation reveal several general features, for example, immune tolerance and invasion.	Significant increases in *RUNX3* mRNA expression levels were reported among female smokers relative to nonsmoking women. *RUNX3* has been established to be fundamental for promoting Th1 phenotypes through IL-4 repression.	[58]
LINE-1-elements	Hypo-methylation	Prevents activation of the placental mineralocorticoid receptor. Associated with cardiovascular disease and with risk factors for both cardiovascular disease and preterm birth.	Significant decreases in LINE-1 element methylation levels were observed in placentas during the third trimester relative to the first trimester. Hypomethylation of LINE-1 elements has been associated to pathological processes, as well as tumorigenesis, abnormal placental function, birth defects, aging, and chronic diseases.	[58]
*IGF-1*	Hyper-methylation	Involved in placental formation and fetal growth; associated with increased *DNMT1* expression.	Maternal IGF-1 levels are negatively correlated with pregnancies complicated by pre-eclampsia. Higher maternal IGF-1 concentrations have been reported.	[60]
*VHL*	Hyper-methylation	Codes for a tumor suppressor protein that is critical for normal placental development.	Gestation in patients with VHL disease stimulates cerebellar hemangioblastoma progression and affects the high VHL disorder-related pregnancy complication rate.	[61]
*RASSF1A*	Hyper-methylation	Activated by promoter hyper-methylation in many tumor types.	Increased levels in the placenta in complicated pregnancies.	[62]

Abbreviations: *11β-HSD*: L11β-Hydroxysteroid dehydrogenase type2, *RUNX3*: RUNX Family Transcription Factor 3, *IGF-1*: Insulin-like growth factor 1, *VHL*: von Hippel-Lindau tumor suppressor and *RASSF1A*: Ras association domain family member 1.

**Table 2 ijms-23-12698-t002:** DNA methylation related to pregnancy maintenance and changes in placental function.

Genes	DNA Methylation Changes	Mechanism of Genes	References
LINE-1-elements, *RUNX3*, *HSD11B2*	Hypo-methylation	Inhibits the activation of mineralocorticoid receptor in placenta.	[58]
*IGF-1*	No difference	Formation of placenta and growth of fetus.	[60]
*DDAH1*	Hyper-methylation	Contribute nitric oxide generation.	[57]
*VHL*	Hyper-methylation	Proper placental development.	[61]
*TERT*	Hypo-methylated	Reverse transcriptase activity.	[105]
*ADORA2B*	Hyper-methylation	Hyper-methylation of this gene associated with hypoxia and PE and sensitive to atmospheric pollutants.	[105]
*CALCA*	Hyper-methylation	Ca++ regulation in placenta.	
*AGT*	Hypo-methylation	Produces angiotensinogen.	[57]
*MMP9*	Hyper-methylation	Trophoblast cell migration	[91]
*DNMT1*, *DNMT3A*	Hypo-methylated	DNA methyltransferases.	[106]
*SPESP1*	Hyper-methylated	Need for successful fertilization.	[80]

Abbreviations: *DDAH1*: Dimethylarginine dimethylaminohydrolase 1, *TERT*: Telomerase reverse transcriptase, *ADORA2B*: Adenosine A2b receptor, *CALCA*: Calcitonin related polypeptide alpha, *AGT*: Angiotensinogen, *MMP9*: Matrix metallopeptidase 9 and *SPESP1*: Sperm equatorial segment protein 1.

**Table 3 ijms-23-12698-t003:** The effects of sex hormones on cells of immune system.

Sex Hormone	DNA Methylation Changes	Target Gene	Biological Effect	References
17-β-estradiol—high concentration	A crucial element in the passive and active DNA demethylation activities both on the DNA and histones.	T helper cell	Amplifies Th2 responses: raises IL-4, IL-10, TGF-β, promotes IRF-1, inhibits TNFα and IL-17.	[122,123,124]
		Treg cell	Promotes differentiation and activity: stimulates FoxP3 and PD-1.	[125,126]
		Nk cell	Declines activity: raises IL-10, TGF-β.	[123,127]
		B cell	Amplifies survival of autoreactive B cells: reduces BCR, increases CD22.	[122]
		Macrophages	Decreases activity: decreases IL-6, IL-1β and TNFα, raises IL-10.	[123,128]
		Dendritic cells	Reduces activity: reduces IL-6, IL-1β and TNFα.	[122,123]
Progesterone	DNA methylation status is still not yet confirmed.	T helper cell	Promotes Th2 response: increases IL-4, IL-10, TGF-β and decreases IFNγ and IL-12. Reduces Th1 and Th17 responses: decreases IFNγ, IL-12, and IL-17.	[122,129,130]
		NK cells	Reduces activity: reduces INFγ.	[122,130]
		B cell	Decreases class-switch recombination and T cells.	[122,129,130]
		Macrophages	Lowers activity: reduces nitric oxide production, TNFα, and FcγR.	[130]
		Dendritic cells	Decreases activity: decreases TLR-mediated IFNα production.	[131]
PSG1a	Highly expressed in myoblasts and strongly downregulated after differentiation.		Enhances IL-10 and TGF-β production.	[132,133]

## Data Availability

Not applicable.

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
