# Peer review of "Epigenetics and Pregnancy: Conditional Snapshot or Rolling Event"

_ijms, 2022, doi:10.3390/ijms232012698_

Round 1
Reviewer 1 Report (New Reviewer)
The manuscript titled “Epigenetics and Pregnancy: Conditional Snapshot or Rolling Event” by Andrawus et al. did an exhaustive review of the literature on epigenetic dynamics during pregnancy, mostly the ones related to DNA methylation. This manuscript provided an interesting summary of the relationship between epigenetics, pregnancy, and pregnancy-related disorders. Overall, this is a well-written manuscript, and I find the topic very useful and attractive to both experts and the general audience. I only have a couple of suggestions which will hopefully help improve this manuscript.
1. Some relevant articles may be of some help to improve the review: https://pubmed.ncbi.nlm.nih.gov/23689534/; https://pubmed.ncbi.nlm.nih.gov/29235940/; https://pubmed.ncbi.nlm.nih.gov/29385276/. I found them particularly useful in the discussion of epigenetic changes in pregnancy from the perspective of stress.
2. It is better to include more content on the other components of epigenetic modification, such as histone modification.
3. 3.2; It is important to note that cellular senescence and epigenetic aging (as shown by the clock) do not directly correspond to each other. i.e. senescent cells may not show accelerated epigenetic aging, while epigenetically older cells or tissues may not have increased senescence hallmarks. I suggest that this be discussed in this section.
Minor issues:
4. Page 1. “Epigenetics modification assumed to be a consequence of changes in an organism’s environment that result in fixed and permanent changes in biological processes” The grammar and the definition sound confusing to me. It is a bit hard for me to understand what exactly the “fixed and permanent changes” define in this sentence, as epigenetic modifications such as DNA methylation can be reversible.
5. Consider reducing the blank spaces surrounding the figures.
6. Figure 4 seems to be flattened.
7. Figure 7: Texts in the figure are not in the middle of the box.
Author Response
October 11, 2022
Dear Editor
We are pleased to submit our revised manuscript entitled “Epigenetics and pregnancy: conditional snapshot or rolling event” (Manuscript ID- ijms-1837533) for consideration of publication in international journal of molecular science.
We would like to thank the reviewers for careful and thorough reading of this manuscript and for the comments and suggestions (have carefully addressed (enclosed)) which, help to improve the quality of this manuscript.
All changes in the manuscript are presented in track changes.
Sincerely
Gil Atzmon, Ph.D.
Associate Professor of Epigenetics and Genomics of Aging/Longevity
University of Haifa
Reviewer 1:
Major comments:
Comment: Some relevant articles may be of some help to improve the review: https://pubmed.ncbi.nlm.nih.gov/23689534/; https://pubmed.ncbi.nlm.nih.gov/29235940/; https://pubmed.ncbi.nlm.nih.gov/29385276/. I found them particularly useful in the discussion of epigenetic changes in pregnancy from the perspective of stress.
Response: Thank you for pointing out that references. These references have been incorporated into the text.
Comment: It is better to include more content on the other components of epigenetic modification, such as histone modification.
Response: As suggested by the reviewer, we include the most epigenetic modification studied such as histone modification and non-coding RNA in our manuscript. Please see page 6-7, lines 288-333.
Comment: 3.2; It is important to note that cellular senescence and epigenetic aging (as shown by the clock) do not directly correspond to each other. i.e. senescent cells may not show accelerated epigenetic aging, while epigenetically older cells or tissues may not have increased senescence hallmarks. I suggest that this be discussed in this section.
Response: thank to the reviewer for pointing this out. We discussed this point in the mentioned section. "Cell senescence is part of the life cycle of all living organisms, including the placenta. It too is formed, grows, performs its multiple functions (e.g., endocrine regulation and nourishment of the fetus), and ages. Cellular senescence results from progressive DNA damage. Which handles the development of the cell’s senescent phenotype and is determined by intrinsic or extrinsic causes. Cell aging may lead to apoptosis, resulting in the aging first of tissues and subsequently organs. Age estimates (epigenetic clocks) based on DNA methylation of epigenetic age, which in most of the human population is like chronological age. Based on the observation that methylation levels of various subsets of CpG sites throughout the genome predictably change over chrono-logical age, the epigenetic aging clocks were innovated. The resulting age estimate by this clock is referred to as epigenetic age or, more precisely, DNAm age. The difference between epigenetic age and chronological age, which reflects the rate of epigenetic aging, carries biological significance: increased epigenetic aging is associated with many age-related pathologies and conditions. As described in Giller el.al study that pregnancy is a model for aging and induces age-related diseases such as diabetes and heart disease.
An accelerated senescence-like state may play a mechanistic role in the patho-physiology of PE supported by published studies showing abnormal senescence and in-creased senescence in PE placentas. It showed that once established, senescence can spread from cell to cell, suggesting that placental senescence may lead to the senes-cence of distant maternal organs and cells. The study of Poganik et.al assessed the analysis of humans and mice conducted pregnancy induces a reversible increase in biological age, peaking around delivery and resolving post-partum
Minor comments:
Comment: Page 1. “Epigenetics modification assumed to be a consequence of changes in an organism’s environment that result in fixed and permanent changes in biological processes” The grammar and the definition sound confusing to me. It is a bit hard for me to understand what exactly the “fixed and permanent changes” define in this sentence, as epigenetic modifications such as DNA methylation can be reversible.
Response: As suggested by the reviewer, we extensively re-wrote this sentence. "Epigenetics modification assumed to be a consequence of changes in an organism’s environment that causing changes in biological processes that control heritable changes in gene expression".
Comment: Consider reducing the blank spaces surrounding the figures.
Response: Edited as suggested.
Comment: Figure 4 seems to be flattened.
Response: This is now revised.
Comment: Figure 7: Texts in the figure are not in the middle of the box.
Response: Thanks for the comments. These are now revised.
Reviewer 2 Report (New Reviewer)
In their work, Andrawus et al. reviewed the association between epigenetic changes during pregnancy and health outcomes in mother and offspring, including gestational diabetes, pre-eclampsia, maternal depression, and others. While this manuscript provides comprehensive overview of studies assessing the role of epigenetics in development of pregnancy-related disorders, its value can be further increased by improvement of its structure and expansion of its content by including data on epigenetic changes associated with the healthy pregnancy process and embryogenic development.
Major points:
1) The structure of the manuscript is not clear and sometimes complicates reading process. For example, chapter 2 consists of several subchapters, including “Epigenetic changes in placenta” where some examples of epigenetic changes in placenta during embryonic development are provided. Another subchapter of the same chapter (“The epigenetics of pre-eclampsia”) reviews epigenetic changes characterizing one of pregnancy-associated disorders. It is not clear why these subchapters have been combined, considering that other disorders are overviewed in separate chapters. The logic and structure of the paper may be improved by first (i) providing the information about natural epigenetic reorganization during pregnancy and embryogenesis processes, followed by (ii) epigenetic changes characterizing maternal pregnancy-associated disorders and (iii) epigenetic changes during pregnancy that affect offspring health later in life.
2) The manuscript can be significantly improved if authors provide additional overview of natural epigenetic changes that happen during gestation in mother cells as well as in the embryo during its development. In particular, recently it was shown that biological age of mothers may be significantly perturbed during pregnancy and restored afterwards (Poganik et al. 2022. Biological age is increased by stress and restored upon recovery. bioRxiv). Overview of natural changes during pregnancy would be a good baseline for the subsequent examination of various pregnancy-associated disorders and their mechanisms at the level of epigenetic regulation.
Minor points:
1) When certain cellular mechanisms of damage are discussed in context of their association with pregnancy-related disorders (e.g., pre-exlampsia), it would be important to specify if there is any evidence that observed associations are causal (if they can indeed induce certain disorders). E.g., epigenetic changes in mtDNA discussed in subchapter 3.1 may play a causal role in development of PE or may be just biomarkers, being a consequence of damage introduced by other factors associated with PE.
2) In some cases, reference in the text is provided not as an index but as a title in brackets (e.g. “Oxidative stress, placental ageing-related pathologies and adverse pregnancy outcomes” in subchapter 3.2).
3) Current version of the manuscript has some typos, they should be corrected (e.g., “Enviromrntal condition” on Figure 3).
Author Response
October 11, 2022
Dear Editor
We are pleased to submit our revised manuscript entitled “Epigenetics and pregnancy: conditional snapshot or rolling event” (Manuscript ID- ijms-1837533) for consideration of publication in international journal of molecular science.
We would like to thank the reviewers for careful and thorough reading of this manuscript and for the comments and suggestions (have carefully addressed (enclosed)) which, help to improve the quality of this manuscript.
All changes in the manuscript are presented in track changes.
Sincerely
Gil Atzmon, Ph.D.
Associate Professor of Epigenetics and Genomics of Aging/Longevity
University of Haifa
Reviewer 2:
Major points:
Comment: The structure of the manuscript is not clear and sometimes complicates reading process. For example, chapter 2 consists of several subchapters, including “Epigenetic changes in placenta” where some examples of epigenetic changes in placenta during embryonic development are provided. Another subchapter of the same chapter (“The epigenetics of pre-eclampsia”) reviews epigenetic changes characterizing one of pregnancy-associated disorders. It is not clear why these subchapters have been combined, considering that other disorders are overviewed in separate chapters. The logic and structure of the paper may be improved by first (i) providing the information about natural epigenetic reorganization during pregnancy and embryogenesis processes, followed by (ii) epigenetic changes characterizing maternal pregnancy-associated disorders and (iii) epigenetic changes during pregnancy that affect offspring health later in life.
Response: As suggested by the reviewer, we have now changed the text structure accordingly.
Comment: The manuscript can be significantly improved if authors provide additional overview of natural epigenetic changes that happen during gestation in mother cells as well as in the embryo during its development. In particular, recently it was shown that biological age of mothers may be significantly perturbed during pregnancy and restored afterwards (Poganik et al. 2022. Biological age is increased by stress and restored upon recovery. bioRxiv). Overview of natural changes during pregnancy would be a good baseline for the subsequent examination of various pregnancy-associated disorders and their mechanisms at the level of epigenetic regulation.
Response: we thank the reviewer for this suggested. This important point was addressed in the revised version of the text. We also include the suggested reference. Please see page 7, lines 335-361 and page 17, lines 670-672.
Minor points:
Comment: When certain cellular mechanisms of damage are discussed in context of their association with pregnancy-related disorders (e.g., pre-exlampsia), it would be important to specify if there is any evidence that observed associations are causal (if they can indeed induce certain disorders). E.g., epigenetic changes in mtDNA discussed in subchapter 3.1 may play a causal role in development of PE or may be just biomarkers, being a consequence of damage introduced by other factors associated with PE.
Response: Edited as suggested. Please see page 14, lines 542-546.
Comment: In some cases, reference in the text is provided not as an index but as a title in brackets (e.g. “Oxidative stress, placental ageing-related pathologies and adverse pregnancy outcomes” in subchapter 3.2).
Response: Thanks for the comments. This comment was addressed.
Comment: Current version of the manuscript has some typos, they should be corrected (e.g., “Enviromrntal condition” on Figure 3).
Response: Edited as suggested.
This manuscript is a resubmission of an earlier submission. The following is a list of the peer review reports and author responses from that submission.
Round 1
Reviewer 1 Report
Overall, this review tackles several aspects that links epigenetic changes to pregnancy outcomes. It is easy to follow and touches the major of the fields that influence epigenetic modification during pregnancy. Even though, some points are treated superficially, just because the review has the ambition of offering a wide overview of the various aspects of pregnancy that might be influenced by epigenetic changes.
There are some imprecisions.
Abstract, page 1, line 12 change “gene alteration” with “modification of gene expression”
Keywords, page 1, line 21 change “pregnany” with “pregnancy”
Page 2 Figure 1 change “millutes” with “mellitus”
Page 2, line 52 The Authors should make an effort and find better citations.
Page 2, line 57 The Authors write “clusters of CpG sites are often methylated”. This is wrong. It must be corrected.
Page 2, Paragraph 1.1 The Authors should also describe TET proteins, which are part of the demethylation process.
Page 4 Line 143 change “sem-alleogenic” with “semi-allogeneic”
Page 5, Figure 3 the Figure legend is not clear. It connects concepts that are not consistent with the figure.
Page 6, line 204. The Authors cite Figure 3 referring to the IGF2/H19 locus, but the concept is not depicted in Figure 3. The Authors describe this locus is Figure 4, but also in this case the Figure Is not correct. They should provide a better description. The legend of Figure 4 should be corrected. Enhancers are not present in the figure.
Page 7 Line 228 The authors write “Kamrani et al. reported that the hyper-methylation of several genes plays a role in the development of PE”. The concept should be expanded and explained.
Page 7 Table 1 columns should be formatted according to the text amount, i.e. the pregnancy effects column should be widened.
Page 8-9 lines 258-261 The Authors states “The epigenetic regulation of mitochondrial function is an area of active research, and three key mechanisms have been reported to regulate gene expression within mitochondria including DNA methylation, non-coding RNAs, and post-translational changes in nucleoid related proteins”. The Authors should make some examples or check if this is confirmed.
Page 9, line 274 The Authors states “Mitochondria certainly contain the machinery required to epigenetically modify mtDNA”. This concepshould be explained given the characteristics of mitochondria.
Page 9, lines 282-287 The Authors states “the study of mtDNA methylation inheritance and its effects on the disease risk occurrence could be of significant benefit (Mitochondrial epigenetics and environmental exposure). Alterations of mtDNA methylation have been associated with multifactorial diseases such as cardiovascular diseases (CVDs) and obesity, as well as with environmental exposures including pollutants and nutrition.” This is an overstatement. The levels of methylation of DNA the CpG and non CpG mitochondrial DNA methylation is controversial. Caution must be used These are two references that should be read
Bicci I, Calabrese C, Golder ZJ, Gomez-Duran A, Chinnery PF. Single-molecule mitochondrial DNA sequencing shows no evidence of CpG methylation in human cells and tissues. Nucleic Acids Res. 2021 49:12757-12768. Morris MJ, Hesson LB, Youngson NA. Non-CpG methylation biases bisulphite PCR towards low or unmethylated mitochondrial DNA: recommendations for the field. Environ Epigenet. 2020 6:dvaa001
Page 9, lines 282-287 The Authors states “Methylation levels of human placental DNA increase significantly with gestational age, perhaps reflecting a gradual loss of cellular plasticity usually associated with higher levels of methylation of DNA” but this regards the levels of methylation of DNA
Page 10 Figure 5 is incorrect:. Mitochondria cannot be associated with telomere length or CpG methylation
Page 10, Missing references for the sentences lines 320 “Many studies have explored the effects of maternal dyslipidemia on the epigenetic aging of the placenta” and 324 “socio-demographic parameters.” References are missing and “Oxidative stress induces the activation of repair pathways, inhibits cellular proliferation through the induction of senescence or transient cell cycle arrest, and can drive apoptotic cell death”
Page 11, Missing references for Franco et al line 352, Birton-Shental et al line 359 and Polack et al line 362
Page 12 line line 401 the number of the reference, 66, for Kang et al is missing
Page 12, lines 434-435 The Authors states “Evidence suggests that maternal stress during pregnancy induces the hyper-methylation of HSD11B2in the placenta.”, but the reference is missing.
Page 13, References 71, 75, 76,77 are not pertinent.
Page 14 The reference to Figure 2 should be change. It could be Figure 6 or Table 2.
Page 15 table 3 is not mentioned in the text.
Page 16 Missing references for the sentences lines 519-521 “Elevated autophagic activity in the placenta during pregnancy has previously been shown to be involved in the pathophysiology of PE, with evidence of an inverse association between glucose and placental autophagy.”
Page 16, Missing references for Finer et al line 521
Page 16 line 528 reference 94 is not pertinent with autophagy
references 95-97 are pertinent with authophagy but not pertinent with the sentence “The differential methylation and expression of these pathways may thus be indicative of numerous physiological changes occurring in the hyperglycaemic placenta, which is consistent with a multifactorial disease model.
Page 16 lines 549-551The Authors should reformulate the sencence “In the winter and spring of 1944 after a railway strike, the German occupation limited rations such that people, including pregnant women, in the western region of The Netherlands, including Amsterdam, received as little as 400–800 calories per day.”
Page 16 lines 558-560 Missing reference for the sentence “Future research has the potential to offer additional insight regarding the ability of prenatal exposure to shape disease risk for multiple generations through epigenetic mechanisms.
Page 17 lines 577-579 Missing reference for the sentence “ Feeding mice a diet rich in methyl donors can influence their coat color, body weight, and the health of their progeny.”
Overall the bibliography should be refined. Other references incorrect, for example Anderson et al at lane 36 and Wang et al at lane 472.
In my opinion, at lane 55, the therapeutic or preventive strategies should be deepened.
Another aspect to correct is the denomination of figure and tables:
Figures 1, 2, 5, 6 and table 3 are not cited in the text.
At lane 204 “Figure 3” should be substituted with “Figure 4”.
At lane 496 “Figure 2” should be substituted with “Table 2”.
At lane 584 “Figure 6” should be substituted with “Figure 7”.
Author Response
August 01, 2022
Dear Editor
We are pleased to submit our revised manuscript entitled “Epigenetics and pregnancy: conditional snapshot or rolling event” (Manuscript ID- ijms-1837533) for consideration of publication in international journal of molecular science.
We would like to thank the reviewer for careful and thorough reading of this manuscript and for the thoughtful and insightful comments and constructive suggestions (have carefully addressed (enclosed)) which, help to improve the quality of this manuscript.
Sincerely
Gil Atzmon, Ph.D.
Associate Professor of Epigenetics and Genomics of Aging/Longevity
University of Haifa
Reviewer 1:
Reviewer Comment: Abstract, page 1, line 12 change “gene alteration” with “modification of gene expression”
Keywords, page 1, line 21 change “pregnany” with “pregnancy”
Page 2 Figure 1 change “millutes” with “mellitus”
Page 2, line 52 The Authors should make an effort and find better citations.
Page 2, line 57 The Authors write “clusters of CpG sites are often methylated”. This is wrong. It must be corrected.
Page 4 Line 143 change “sem-alleogenic” with “semi-allogeneic”.
Response: We appreciate the Reviewer's comment. We have checked and corrected the typos and errors.
Reviewer Comment: Page 2, Paragraph 1.1 the Authors should also describe TET proteins, which are part of the demethylation process.
Response: We thank the reviewer for the comment. We have added a description of TET proteins. Please see Page 3, lines: 61-72.
Reviewer Comment: Page 5, Figure 3 the Figure legend is not clear. It connects concepts that are not consistent with the figure.
Response: We thank the reviewer for this comment, the figure legend is corrected. “Figure 3. Environmental and genetic risk factors promote changes in DNA methylation. Candidate genes that are involved in maternal adaptation to pregnancy based on their altered methylation profiles may be affected by epigenetic reprograming process.”
Reviewer Comment: Page 6, line 204. The Authors cite Figure 3 referring to the IGF2/H19 locus, but the concept is not depicted in Figure 3. The Authors describe this locus is Figure 4, but also in this case the Figure is not correct. They should provide a better description. The legend of Figure 4 should be corrected. Enhancers are not present in the figure.
Response: Thank you. We have corrected the number and the description of the figure.
Reviewer Comment: Page 7 Line 228 the authors write “Kamrani et al. reported that the hyper-methylation of several genes plays a role in the development of PE”. The concept should be expanded and explained.
Response: As suggested by the reviewer, we have added some examples of several genes that plays role in the development of PE. Page 9, lines: 271-282. In addition, table 1 includes the function of many genes associated with the development of PE.
Reviewer Comment: Page 9, lines 282-287 The Authors states “the study of mtDNA methylation inheritance and its effects on the disease risk occurrence could be of significant benefit (Mitochondrial epigenetics and environmental exposure). Alterations of mtDNA methylation have been associated with multifactorial diseases such as cardiovascular diseases (CVDs) and obesity, as well as with environmental exposures including pollutants and nutrition.” This is an overstatement. The levels of methylation of DNA the CpG and non CpG mitochondrial DNA methylation is controversial. Caution must be used these are two references that should be read.
Bicci I, Calabrese C, Golder ZJ, Gomez-Duran A, Chinnery PF. Single-molecule mitochondrial DNA sequencing shows no evidence of CpG methylation in human cells and tissues. Nucleic Acids Res. 2021 49:12757-12768.
Morris MJ, Hesson LB, Youngson NA. Non-CpG methylation biases bisulphite PCR towards low or unmethylated mitochondrial DNA: recommendations for the field. Environ Epigenet. 2020 6:dvaa001.
Response: We agree with the reviewer. Thank you for pointing out that references. We used these references in our document.
Reviewer Comment: Page 9, lines 282-287 The Authors states “Methylation levels of human placental DNA increase significantly with gestational age, perhaps reflecting a gradual loss of cellular plasticity usually associated with higher levels of methylation of DNA” but this regards the levels of methylation of DNA.
Response: In humans the placenta is also regarded to be the most hypo-methylated tissue, although recent evidence suggests that differences in tissue-specific methylation patterns are perhaps more notable than differences in global 5mC levels. Novakovic et.al shows a decreasing in gene expression with increasing methylation, highlighting the functional relevance of DNA methylation in trophoblast cell.
Reviewer Comment: Page 10 Figure 5 is incorrect: Mitochondria cannot be associated with telomere length or CpG methylation.
Response: Thank you for pointing out that. The figure edited as suggested.
Reviewer Comment: Page 10, Missing references for the sentences lines 320 “Many studies have explored the effects of maternal dyslipidemia on the epigenetic aging of the placenta” and 324 “socio-demographic parameters.” References are missing and “Oxidative stress induces the activation of repair pathways, inhibits cellular proliferation through the induction of senescence or transient cell cycle arrest, and can drive apoptotic cell death”.
Page 11, Missing references for Franco et al line 352, Birton-Shental et al line 359 and Polack et al line 362
Page 12 line 401 the number of the reference, 66, for Kang et al is missing
Page 12, lines 434-435 The Authors states “Evidence suggests that maternal stress during pregnancy induces the hyper-methylation of HSD11B2in the placenta.”, but the reference is missing.
Page 13, References 71, 75, 76, and 77 are not pertinent.
Page 14 the reference to Figure 2 should be change. It could be Figure 6 or Table 2.
Page 16 Missing references for the sentences lines 519-521 “Elevated autophagy activity in the placenta during pregnancy has previously been shown to be involved in the pathophysiology of PE, with evidence of an inverse association between glucose and placental autophagy.”
Page 16, Missing references for Finer et al line 521
Page 16 line 528 reference 94 is not pertinent with autophagy.
References 95-97 are pertinent with autophagy but not pertinent with the sentence “The differential methylation and expression of these pathways may thus be indicative of numerous physiological changes occurring in the hyperglycemic placenta, which is consistent with a multifactorial disease model.
Page 16 lines 549-551The Authors should reformulate the sentence “In the winter and spring of 1944 after a railway strike, the German occupation limited rations such that people, including pregnant women, in the western region of The Netherlands, including Amsterdam, received as little as 400–800 calories per day.”
Page 16 lines 558-560 Missing reference for the sentence “Future research has the potential to offer additional insight regarding the ability of prenatal exposure to shape disease risk for multiple generations through epigenetic mechanisms.
Page 17 lines 577-579 Missing reference for the sentence “Feeding mice a diet rich in methyl donors can influence their coat color, body weight, and the health of their progeny.
Response: Thank you for this comments. We included the following reference in the text respectively.
Reviewer Comment: Page 15 table 3 is not mentioned in the text.
Response: Thank you for this comment. Table 3 is mentioned in the text (paragraph 4, Page 15).
Reviewer Comment: In my opinion, at lane 55, the therapeutic or preventive strategies should be deepened.
Response: We agree with Reviewer for this recommendation. The epigenetic changes are reversible and consequently are potential contributors to health and disease. Epigenetic biomarkers are emerging as tools for the screening and early detection of various diseases, for prognostic and treatment monitoring, and for predicting future risk of disease development. In clinical practice, this will contribute to the development of in vitro diagnostic and prognostic tools, which in turn may also lead to advances in personalized medicine.
Reviewer Comment: Figures 1, 2, 5, 6 and table 3 are not cited in the text.
At lane 204 “Figure 3” should be substituted with “Figure 4”.
At lane 496 “Figure 2” should be substituted with “Table 2”.
At lane 584 “Figure 6” should be substituted with “Figure 7”.
Response: These specific amendments will be addressed in the updated manuscript.
** Changes to the text are labeled with red.
We hope these additions and clarifications will meet with your approval, and our manuscript will be considered for publication in International Journal of Molecular Sciences.
Reviewer 2 Report
In this work, Andrawus et al aim to provide an overview of the alterations that epigenetic factors, normally regulating pregnancy and fetal development, undergo under conditions that may eventually lead to the development of pathologies.
The topic is of potential interest to the scientific audience but unfortunately it is poorly presented.
The entire manuscript requires a careful revision by the authors before it can be considered for external review.
The following are my comments/suggestions. Please consider that I provide some examples, but the comments are valid for the entire manuscript.
- The focus of the work must be clearly presented and a logical flow must be followed
- The titles of the paragraphs must be coherent with their content (for example, paragraph 1.1 “Maternal DNA during pregnancy” describes DNA methylation mechanism in general).
- The authors must provide the knowledge to the reader to understand a topic. For example, the role of S-adenosylmethionine in DNA methylation should be introduced before discussing the effects on DNA methylation derived from the assumption of methyl-group donors derived from food.
- Please avoid comments about the importance of new studies in the field at the end of each paragraph but reserve them for the conclusions.
- Please introduce and discuss each topic equally
- Figures must be exemplificative of the content and figure legends must be descriptive of the figures
- Figures and tables must be revised and cited correctly along the manuscript. Now many errors are present in the figures and in the tables (e.g. headings and content of the tables, spelling errors, etc.). Many references to figures and tables are missing or they are wrongly cited.
- Bibliographic references are missing in many parts of the manuscript. Please cite appropriate studies when reporting evidence.
- An extensive editing of English language and style is required.
Minor comments:
- Some sentences are incomplete (e.g. line 295 “When cells finally differentiate.”)
- Please use italics for non-english words (e.g. de novo)
- Use CGI as acronym for CpG island
Author Response
August 01, 2022
Dear Editor
We are pleased to submit our revised manuscript entitled “Epigenetics and pregnancy: conditional snapshot or rolling event” (Manuscript ID- ijms-1837533) for consideration of publication in international journal of molecular science.
We would like to thank the reviewer for careful and thorough reading of this manuscript and for the thoughtful and insightful comments and constructive suggestions (have carefully addressed (enclosed)) which, help to improve the quality of this manuscript.
Sincerely
Gil Atzmon, Ph.D.
Associate Professor of Epigenetics and Genomics of Aging/Longevity
University of Haifa
Reviewer 2:
Reviewer comment: The focus of the work must be clearly presented and a logical flow must be followed.
Response: Thanks for the reviewer constructive comments. The comments are significantly helpful to improve readability of our manuscript, and make the paper more solid. The following presents our point-to-point responses as well as the revision for the manuscript.
Reviewer comment: The titles of the paragraphs must be coherent with their content (for example, paragraph 1.1 “Maternal DNA during pregnancy” describes DNA methylation mechanism in general).
Response: Thanks for this comment, this is now revised as the reviewer suggested in the manuscript.
Reviewer comment: The authors must provide the knowledge to the reader to understand a topic. For example, the role of S-adenosyl methionine in DNA methylation should be introduced before discussing the effects on DNA methylation derived from the assumption of methyl-group donors derived from food.
Response: Thanks for the comments. More volume and explanation were made to the relevant parts in the text.
Reviewer comment: Please avoid comments about the importance of new studies in the field at the end of each paragraph but reserve them for the conclusions.
Response: We appreciate the Reviewer's comment and omitted all the comments about the importance of new studies moved to conclusion.
Reviewer comment: Please introduce and discuss each topic equally.
Response: Thank you, we corrected it in the text.
Reviewer comment: Figures must be exemplificative of the content and figure legends must be descriptive of the figures.
Response: Thank you for your comment. We modified the figure according to the reviewer suggestion.
Reviewer comment: Figures and tables must be revised and cited correctly along the manuscript. Now many errors are present in the figures and in the tables (e.g. headings and content of the tables, spelling errors, etc.). Many references to figures and tables are missing or they are wrongly cited.
Response: We have revised the manuscript carefully and tried to avoid any grammar or syntax error.
Reviewer comment: Bibliographic references are missing in many parts of the manuscript. Please cite appropriate studies when reporting evidence.
Response: We thank the reviewer for this comment, we have checked the whole manuscript and added the missing references.
Reviewer comment: An extensive editing of English language and style is required.
Response: We checked the manuscript for English editing and modified the sentences according to the reviewer suggestion. In addition, this report undergo extensive English editing by expert in the field.
Reviewer comment: Some sentences are incomplete (e.g. line 295 “When cells finally differentiate.”).
Response: Thank the reviewer for pointing out this error, we have checked and corrected these errors throughout the manuscript.
Reviewer comment: Please use italics for non-English words (e.g. de novo).
Response: Thank the reviewer for pointing out this matter, such items were edited as suggested.
Reviewer comment: Use CGI as acronym for CpG Island.
Response: As suggested by the reviewer, we used CGI as an acronym.
** Changes to the text are labeled with red.
We hope these additions and clarifications will meet with your approval, and our manuscript will be considered for publication in International Journal of Molecular Sciences.
Round 2
Reviewer 1 Report
This is the third revision of the review titled “Epigenetics and Pregnancy: Conditional Snapshot or Rolling Event” by Andrawus et al.
The Authors made an effort to give a general view on epigenetics in pregnancy. The topic is interesting. The main problem regards the selection of the published literature. In this respect the review suffers of some naivete. The study of epigenetics in humans and in adult cells is extremely challenging. This might bring over-interpretation o misinterpretation of results. This possibility should be of concern for the Authors.
Since its first version the review has significantly improved, even though there are several aspects that need to be corrected or revised.
In some points the text is confusing and/or not consequential, as well the English language need some editing.
Here is my point-to-point review
The abstract is not clear. It must be re-written.
- Page 1 line 30-31 There are more histone modifications. The authors should use as reference a general review on histone modifications. I suggest Millán-Zambrano G, Burton A, Bannister AJ, Schneider R. Histone post-translational modifications - cause and consequence of genome function. Nat Rev Genet. 2022 23:563-580.
- Page 2 line 59-61 “Gene expression also regulated de-methylation of DNA by TET protein (Ten-Eleven Translocation proteins) which oxidize 5-methylcytosines.” The sentence is not clear. It must be re-written.
- Figure 1 No DNA methylation is represented in this figure
- Paragraph 2. Pregnancy and epigenetic changes. In all this part the type of cells on which the studies were conducted should be reported.
- Page 5, line 152 The Authors state “Pregnancy-related risk factors can influence maternal DNA methylation patterns, potentially increasing clinical risk and rendering mothers susceptible to long-term consequences. “ This is a very general statement, however in the same individual the methylation status of a genomic region can change depending on the tissue analyzed. This is a limitation of epigenetic studies, because results might not reflect the epigenetic status of the tissue, which is critical for the condition considered in the study. In other words, as an example, does the methylation status in lymphocytes reflect the methylation status in liver or in other relevant tissues when epigenetic changes related to nutrition are investigated? The Authors should comment on this point.
- Page 9, lines 307-317 The Author reported “Mammalian mtDNA is a circular double-stranded molecule, and encodes only 37 genes, 13 of which are respiratory chain subunits and 24 being RNA components, such as tRNAs and rRNAs. The mitochondria is the power of the cell and is involved in many cellular processes, including oxidative phosphorylation and ATP production, b-fatty acid oxidation, the citric acid cycle, and regulation of apoptosis. Mitochondrial epigenetics is an emerging field that examines how mtDNA methylation may affect not only development but also health and disease later in life. It should also be postulate that, as mitochondria is a major sole maternal inheritance (despite some evidence for mitochondrial sperm inheritance), the study of mtDNA methylation inheritance and its effects on the disease risk occurrence could be of significant benefit (Mitochondrial epigenetics and environmental exposure).” This part should be at the beginning of the section 3.1 Mitochondrial changes during pregnancy.
- Page 9, lines 307-317 The Authors wrote “Some publications supplyed to shift in opinion as they make available technical developments, in addition to information on regional CpG and non-CpG mtDNA methylation, regulation by methyltransferases and relations with disease [61–63].” This sentence is not clear it should be rephrased.
- Page 9, lines 330-333 The Authors wrote “If these changes in oxygen concentration occur suddenly, it may lead to a change in cell signaling and an increase in damage indicating that the placenta is no longer aging chronologically and causing pla cental pathologies [68]. “This sentence is not clear it should be reformulated.
- Page 9, lines 334-337 The Authors wrote “When cells finally differentiate, a special feature of the trophoblast epigenome is the prevalence of large areas, so-called intermediate methylation, which extend across transcriptionally repressed, polycomb-focused regions of the genome. This function of the epigenetic is significance remains unknown.” The Authors should make this sentence clear and refer to the article it is coming from.
- Pages 10-11, lines 367-375 The Authors wrote “Oxidative stress induces the activation of repair pathways, inhibits cellular proliferation through the induction of senescence or transient cell cycle arrest, and can drive apoptotic cell death [74] . Oxidative stress is defined as an imbalance between the generation of reactive oxygen species (ROS) and the ability of antioxidant defense mechanisms to mitigate ROS-induced damage. Pregnancy is intrinsically linked to a state of persistent oxidative stress owing to increased metabolic activity in placental mitochondria and corresponding ROS production. While an appropriate homeostatic balance between ROS generation and antioxidant activity is maintained in normal pregnancies, the disruption of this balance can contribute to oxidative stress.” This part should be reorganized.
- Page 11, lines 376-406 The section 3.3 Telomerase activity and telomere length is not pertinent to the review. It should be removed
- Page 13, lines 503-506. The Authors wrote “The study of Pauwels et.al found that women who’s took a folic acid supplement before and during pregnancy, can assume that was a delay in DNA methylation response, it would have happened before pregnancy [21].” The sentence is not clear. It should be made clear.
- Page 15 Table 3. DNA methylation related to pregnancy maintenance and changes in placenta function. The acronym LINE-1 defines a family of Long INterspersed repetitive Elements. It is NOT a gene. This should be specified along the text.
- Page 16 Section 8. The effects of epigenetic inheritance on pregnancy. In this section there are some redundancies. I suggest to reorganize sentences to avoid repetitions
- Page 16, lines 597-598 The sentence “The hight of the famine in the end of 1944 until first quarter of 1945 official daily rations varied between 400 and 800 calories.” Should be rewritten.
- Page 16, lines 605-607 The sentence “Future research has the potential to offer additional insight regarding the ability of prenatal exposure to shape disease risk for multiple generations through epigenetic mechanisms [125].” should be moved to the end of section 8.
- Page 18. Conclusions lines 660 and following.
In the conclusion I would refer to the experimental difficulties that researchers can encounter in the analyses of this processes, e.g. type of cells that can be analyzed, inter-individual variability, sample size, etc. These should be considered limitations to these population-based epigenetic studies.
Minor comments
- Page 3 line 93. The subject should be before the verb.
- Page 3 line 110. The reference is not listed in the bibliography.
- Page 4 line 159. The reference is not listed in the bibliography.
- Page 5 line 171. The reference is not listed in the bibliography.
- Page 5 line 192 “foundational” must be changed.
- Page 7 Table 1. The column “pregnancy effects” can be widened and that “references” reduced
- Page 8, lines 291 the Authors wrote “Several novel studies have examined...” without references. Please add.
- Page 9, lines 298 the Authors wrote “Mitochondria certainly contain...” without references. Please add.
- Page 9, lines 321 the Authors wrote “Even though the fact that previous studies...” without references. Please add.
- Page 10 Caption of Figure 5 is not clear. Please check.
- Page 12, lines 427-431 the Authors wrote “Notably, placental DNA methylation has been found...” without references. Please add.
- Page 12, line 458. Reference 92 is not pertinent. The correct reference must be inserted.
- Reference 92 refers to the sentence “Prenatal maternal depression can cause covalent epigenetic changes in the DNA of their offspring that are detectable at birth in leukocytes and that may be present in other tissues, suggesting a model wherein systemic epigenetic changes may be involved in lifelong responses to the in utero psychosocial environment.” at lines 461-464
- Page 12, lines 464-466 “Maternal depression during pregnancy has been linked to an increased risk of obstetric complications such as GDM, hypertension, and PE, as well as smoking, increased alcohol consumption, and inadequate nutrition.” The reference is missing.
- Reference 95 is not pertinent.
- Page 13, lines 482-484 “Nutrients can act directly by inhibiting...” Reference should be added.
- Page 13, lines 486-487 “This, in turn, changes the expression of critical genes...” Reference should be added.
- Page 13, lines 492-493 “Other studies...” Reference should be added.
- Page 13, line 494 “Present work...” Substitute with the name of the author “The work by...”
- Page 13, lines 498-500. Check reference 99, if appropriate.
- Page 13 lines 506-509 “Low concentrations of vitamin D....” the reference is missing.
- Page 13, line 512 “most defenseless” should be changed “is more susceptible”
- Page 14, line 527-528 References are missing.
- Page 14, line 535-536 References are missing.
- Page 16, line 568 Is reference 118 pertinent?
- Page 16, lines 581-584 “The role of the developmental and parental environmental exposures in shaping the ultimate metabolic characteristics of offspring has been effectively demonstrated in several human studies of populations exposed to extreme nutrient deficiencies during pregnancy. Reference should be added.
- Page 16, lines 599-602. References are missing
- Page 16, lines 608-610. References are missing
- Page 17, Figure 7 “Enviroment factors” should become “Environmental factors”
Author Response
September 9, 2022
Dear Editor
We are pleased to submit our revised manuscript entitled “Epigenetics and pregnancy: conditional snapshot or rolling event” (Manuscript ID- ijms-1837533) for consideration of publication in international journal of molecular science.
We thank all reviewers for taking time to review our manuscript and for offering many constructive comments, which have contributed toward improving the quality of our paper. We have addressed all of their queries (enclose).
Sincerely
Gil Atzmon, Ph.D.
Associate Professor of Epigenetics and Genomics of Aging/Longevity
University of Haifa
Reviewer 1:
Reviewer Comment: The abstract is not clear. It must be re-written.
Response: Thank you very much for your suggestions, we definitely agree with the reviewers comment. The abstract was significantly revised.
Reviewer Comment: Page 1 line 30-31 there are more histone modifications. The authors should use as reference a general review on histone modifications. I suggest Millán-Zambrano G, Burton A, Bannister AJ, Schneider R. Histone post-translational modifications - cause and consequence of genome function. Nat Rev Genet. 2022 23:563-580.
Response: Thank you for pointing out that references. The text now includes the citation.
“The most common PTM are methylation, acetylation, phosphorylation, and ubiquitination, while sumoylation, ADP-ribosylation, deimination, and proline isomerization are more recent discoveries that have yet to be thoroughly investigated”.
Reviewer Comment: Page 2 line 59-61 “Gene expression also regulated de-methylation of DNA by TET protein (Ten-Eleven Translocation proteins) which oxidize 5-methylcytosines.” The sentence is not clear. It must be re-written.
Response: Thanks for the comments. We have revised the sentence and it’s now reads as follow. “Several studies have suggested that TET protein regulates DNA methylation and gene expression through a variety of mechanisms (Tet family proteins and 5-hydroxymethylcytosine in development and disease)”. The TET proteins have three members, TET1, TET2, and TET3, which are responsible for the conversion of 5-mC to 5-hmC”.
Reviewer Comment: Figure 1 No DNA methylation is represented in this figure.
Response: Thanks for the comments. The description is now revised to “Complications during pregnancy”.
Reviewer Comment: Paragraph 2. Pregnancy and epigenetic changes. In all this part the type of cells on which the studies were conducted should be reported.
Response: Thanks to the reviewer's constructive comments, all types of cells are added, respectively, in the text.
Reviewer Comment: Page 5, line 152 The Authors state “Pregnancy-related risk factors can influence maternal DNA methylation patterns, potentially increasing clinical risk and rendering mothers susceptible to long-term consequences. “ This is a very general statement, however in the same individual the methylation status of a genomic region can change depending on the tissue analyzed. This is a limitation of epigenetic studies, because results might not reflect the epigenetic status of the tissue, which is critical for the condition considered in the study. In other words, as an example, does the methylation status in lymphocytes reflect the methylation status in liver or in other relevant tissues when epigenetic changes related to nutrition are investigated? The Authors should comment on this point.
Response: Thanks for the comment and suggestion. We added more discussions on the drawbacks of this point. “There is a growing need to investigate the genetic and epigenetic factors such as DNAm from minimally invasive maternal tissues such as whole blood, which can help monitor fetal-maternal conditions during pregnancy. Blood provides a unique potential window into an individual's health and phenotype, and hence longitudinal epigenetic profiling of peripheral blood DNA from pregnant women during pregnancy and postpartum may provide improved insights into the physiological changes that occur. This may be used to identify potential biomarkers of pregnancy outcomes. Environmental factors and genetic factors have also been assessed regarding their association with each cell type or tissue. Epigenetic variation is regulated in a tissue- and developmental stage-specific manner. Because of the enormous between-tissue differences in DNA methylation patterns, it is a major challenge to identify informative epigenetic signatures in blood or placenta epigenomes and then confirm that they reflect at least to some extent methylation variation in the target tissues. A study on maternal venous white blood cells identified changes in methylation profiles of the LINE-1 elements, which are typically heavily methylated in normal conditions but are hypo-methylated during cellular stress. The study suggested that hyper-methylation of these elements in early pregnancy may lead to preterm birth. In another word, dynamic DNAm changes regulate the variable expression of the genome in diverse maternal tissues like the uterus, breasts, etc. and response to environmental factors from nulligravid to postpartum state.”
Reviewer Comment: Page 9, lines 307-317 The Author reported “Mammalian mtDNA is a circular double-stranded molecule, and encodes only 37 genes, 13 of which are respiratory chain subunits and 24 being RNA components, such as tRNAs and rRNAs. The mitochondria is the power of the cell and is involved in many cellular processes, including oxidative phosphorylation and ATP production, b-fatty acid oxidation, the citric acid cycle, and regulation of apoptosis. Mitochondrial epigenetics is an emerging field that examines how mtDNA methylation may affect not only development but also health and disease later in life. It should also be postulate that, as mitochondria is a major sole maternal inheritance (despite some evidence for mitochondrial sperm inheritance), the study of mtDNA methylation inheritance and its effects on the disease risk occurrence could be of significant benefit (Mitochondrial epigenetics and environmental exposure).” This part should be at the beginning of the section 3.1 mitochondrial changes during pregnancy.
Response: We agree with the reviewer's comment, thus, we moved this part to the beginning of the paragraph.
Reviewer Comment: Page 9, lines 307-317 The Authors wrote “Some publications supplied to shift in opinion as they make available technical developments, in addition to information on regional CpG and non-CpG mtDNA methylation, regulation by methyltransferases and relations with disease [61–63].” This sentence is not clear it should be rephrased.
Response: Thank the reviewer for pointing this out. Edited as suggested to read as follow.
“Some publications contribute to the shift in this opinion as they supply available technical developments, in addition to information on regional CpG and non-CpG mtDNA methylation, regulation by methyltransferases, and relations with diseases [58–60].”
Reviewer Comment: Page 9, lines 330-333 The Authors wrote “If these changes in oxygen concentration occur suddenly, it may lead to a change in cell signaling and an increase in damage indicating that the placenta is no longer aging chronologically and causing placental pathologies [68]. “This sentence is not clear it should be reformulated.
Response: Thank the reviewer for pointing this out. Edited as suggested.
Reviewer Comment: Page 9, lines 334-337 The Authors wrote “When cells finally differentiate, a special feature of the trophoblast epigenome is the prevalence of large areas, so-called intermediate methylation, which extend across transcriptionally repressed, polycomb-focused regions of the genome. This function of the epigenetic is significance remains unknown.” The Authors should make this sentence clear and refer to the article it is coming from.
Response: We agree with the reviewer’s comment. We made it clearer now and the following references were added.
“Schroeder, D.I.; Blair, J.D.; Lott, P.; Yu, H.O.K.; Hong, D.; Crary, F.; Ashwood, P.; Walker, C.; Korf, I.; Robinson, W.P.; et al. The Human Placenta Methylome. Proc. Natl. Acad. Sci. U. S. A. 2013, 110, doi:10.1073/pnas.1215145110.
Novakovic, B.; Gordon, L.; Wong, N.C.; Moffett, A.; Manuelpillai, U.; Craig, J.M.; Sharkey, A.; Saffery, R. Wide-Ranging DNA Methylation Differences of Primary Trophoblast Cell Populations and Derived Cell Lines: Implications and Opportunities for Understanding Trophoblast Function. Mol. Hum. Reprod. 2011, 17, doi:10.1093/molehr/gar005.
Smith, Z.D.; Shi, J.; Gu, H.; Donaghey, J.; Clement, K.; Cacchiarelli, D.; Gnirke, A.; Michor, F.; Meissner, A. Epigenetic Restriction of Extraembryonic Lineages Mirrors the Somatic Transition to Cancer. Nature 2017, 549, doi:10.1038/nature23891.
Hemberger, M.; Hanna, C.W.; Dean, W. Mechanisms of Early Placental Development in Mouse and Humans. Nat. Rev. Genet. 2020, 21.”
“The maintenance of DNA methylation at sites not associated with known imprinted genes seems to be important for trophoblast development. Human placental DNA methylation levels increase substantially with gestational age, possibly reflecting a gradual loss of cellular plasticity often related to higher DNA methylation levels as cells terminally differentiate. When cells finally differentiate, a special feature of the trophoblast epigenome is the prevalence of large areas, so-called intermediate methylation, which extend across transcriptionally repressed, polycomb-focused regions of the genome. This function of the epigenetic is significance remains unknown”
Reviewer Comment: Pages 10-11, lines 367-375 The Authors wrote “Oxidative stress induces the activation of repair pathways, inhibits cellular proliferation through the induction of senescence or transient cell cycle arrest, and can drive apoptotic cell death [74]. Oxidative stress is defined as an imbalance between the generation of reactive oxygen species (ROS) and the ability of antioxidant defense mechanisms to mitigate ROS-induced damage. Pregnancy is intrinsically linked to a state of persistent oxidative stress owing to increased metabolic activity in placental mitochondria and corresponding ROS production. While an appropriate homeostatic balance between ROS generation and antioxidant activity is maintained in normal pregnancies, the disruption of this balance can contribute to oxidative stress.” This part should be reorganized.
Response: We extensively re-wrote this part, as suggested by the reviewer and it’s read as follow.
“Oxidative stress is recognized as a key factor in the pathogenesis of adverse pregnancy outcomes. The accumulation of oxidative stress causes damage to lipids, proteins and DNA in the placental tissue that induces a form of accelerated ageing. Many studies indicated that oxidative stress and placental ageing play a role in the pathophysiology of ab-normal pregnancies, with a particular emphasis on pregnancy complicated by spontaneous preterm birth, intrauterine growth restriction, pre-eclampsia, pregnancy loss and still-birth. (Oxidative stress, placental ageing-related pathologies and adverse pregnancy out-comes). Even though a physiological balance between ROS and antioxidant activity is maintained in normal pregnancies, an imbalance may increase oxidative stress. The placenta experiences an increased level of oxidative stress in certain pathological pregnancies, particularly those complicated by maternal smoking, gestational diabetes, fetal growth restriction, preeclampsia, and miscarriage”
Reviewer Comment: Page 11, lines 376-406 the section 3.3 Telomerase activity and telomere length is not pertinent to the review. It should be removed.
Response: We omit this part.
Reviewer Comment: Page 13, lines 503-506. The Authors wrote “The study of Pauwels et.al found that women who’s took a folic acid supplement before and during pregnancy, can assume that was a delay in DNA methylation response, it would have happened before pregnancy [21].” The sentence is not clear. It should be made clear.
Response: Thanks for the comments. This sentence was revised and it’s read as follow.
“During pregnancy there is a higher need for methyl-group donors like folate, betaine, choline, and methionine , the study of Pauwels et.al examined whether there is a change in methyl-group donor intake and in the DNA methylation pattern in maternal white blood cells during the course of pregnancy, they found high intake of methionine, choline, and betaine pre-pregnancy, lower in the first 12 weeks and a high intake of these three methyl-group donors in the second and third trimester of pregnancy.”
Reviewer Comment: Page 15 Table 3. DNA methylation related to pregnancy maintenance and changes in placenta function. The acronym LINE-1 defines a family of Long Interspersed repetitive Elements. It is NOT a gene. This should be specified along the text.
Response: We agree with the reviewer’s comment. We specifically cite LINE-1 as a repetitive element in the entire text.
Reviewer Comment: Page 16 Section 8. The effects of epigenetic inheritance on pregnancy. In this section there are some redundancies. I suggest to reorganize sentences to avoid repetitions.
Response: We re-edit this part, as suggested by the reviewer.
Reviewer Comment: Page 16, lines 597-598 the sentence “The height of the famine in the end of 1944 until first quarter of 1945 official daily rations varied between 400 and 800 calories.” Should be rewritten.
Response: We extensively re-wrote this part, as suggested by the reviewer and it’s read as follow.
“The height of the famine was from the end of 1944 until first quarter of 1945. The official daily rations varied between 400 and 800 calories per day at the end of 1944 and in the first month of 1945.”
Reviewer Comment: Page 16, lines 605-607 the sentence “Future research has the potential to offer additional insight regarding the ability of prenatal exposure to shape disease risk for multiple generations through epigenetic mechanisms [125].” should be moved to the end of section 8.
Response: We re-wrote this sentence, as suggested by the reviewer and it’s read as follow.
”It is clear that epigenetic regulation is a key feature of pregnancy and development, but our current understanding is rather piecemeal. Knowledge of such epigenetic mechanisms may be useful in identifying novel biomarkers for pregnancy-related exposure, burden, or disease risk. Such biomarkers may prove essential for the development of p tools for the early identification of risk factors and exposure levels.”
Reviewer Comment: Page 18. Conclusions lines 660 and following.
In the conclusion I would refer to the experimental difficulties that researchers can encounter in the analyses of this processes, e.g. type of cells that can be analyzed, inter-individual variability, sample size, etc. These should be considered limitations to these population-based epigenetic studies.
Response: Based on the reviewer's comment, additional details on experimental difficulties that researchers can encounter in the analysis of these processes were added.
“There is a general limitation to the most studies based on whole blood DNAm. Whole blood is composed of multiple cell types, and each cell type contributes to the CpG locus-specific DNA methylation signal [85]. Research on DNAm in whole blood conducted to date has not yet fully addressed the issue of cellular heterogeneity. Although various methods exist for adjusting for cellular heterogeneity, they commonly use reference methylation data, based on which the sample cell proportions are estimated [140]. DNA-based deconvolution of selected cell type-specific differentially methylated regions (DMRs) from sorted and purified cell populations of individual subjects has yet to be investigated. Moreover, during pregnancy, maternal blood also contains fetal genetic materials, and thus there remains a necessity to account for both to exclusively parse the maternal component. Another limitation of pregnancy researchers is the sample size, these limitations avoid the findings are generalized. Gene-specific or genome wide DNA methylation analyses may provide more insight into mechanisms underlying the processes prompting pre-term birth, while such technologies exist, they continue to be costly for cohort studies with large sample sizes [27]. Lastly, the relatively high socioeconomic status of the cohort may limit the generalizability of the findings [141].”
Minor comments
Reviewer Comment: Page 3 line 93. The subject should be before the verb.
Response: Edited as suggested.
Reviewer Comment: Page 3 line 110. The reference is not listed in the bibliography.
Response: The text now includes the following reference. “Gruzieva, O.; Merid, S.K.; Chen, S.; Mukherjee, N.; Hedman, A.M.; Almqvist, C.; Andolf, E.; Jiang, Y.; Kere, J.; Scheynius, A.; et al. DNA Methylation Trajectories During Pregnancy. Epigenetics Insights 2019, 12.”
Reviewer Comment: Page 4 line 159. The reference is not listed in the bibliography.
Response: The text now includes the following reference. “White, W.M.; Sun, Z.; Borowski, K.S.; Brost, B.C.; Davies, N.P.; Rose, C.H.; Garovic, V.D. Preeclampsia/Eclampsia Candidate Genes Show Altered Methylation in Maternal Leukocytes of Preeclamptic Women at the Time of Delivery. Hypertens. Pregnancy 2016, 35.”
Reviewer Comment: Page 5 line 171. The reference is not listed in the bibliography.
Response: The text now includes the following reference. “Babenko, O.; Kovalchuk, I.; Metz, G.A.S. Stress-Induced Perinatal and Transgenerational Epigenetic Programming of Brain Development and Mental Health. Neurosci. Biobehav. Rev. 2015, 48.”
Reviewer Comment: Page 5 line 192 “foundational” must be changed.
Response: the word was changed to “earliest”.
Reviewer Comment: Page 7 Table 1. The column “pregnancy effects” can be widened and that “references” reduced.
Response: Thanks for the comment. This point was fixed.
Reviewer Comment: Page 8, lines 291 the Authors wrote “Several novel studies have examined...” without references. Please add.
Response: The text now includes the following reference. “Pirini, F.; Guida, E.; Lawson, F.; Mancinelli, A.; Guerrero-Preston, R. Nuclear and Mitochondrial DNA Alterations in Newborns with Prenatal Exposure to Cigarette Smoke. Int. J. Environ. Res. Public Health 2015, 12.”
Reviewer Comment: Page 9, lines 298 the Authors wrote “Mitochondria certainly contain...” without references. Please add.
Response: The text now includes the following reference. “Armstrong, D.A.; Green, B.B.; Blair, B.A.; Guerin, D.J.; Litzky, J.F.; Chavan, N.R.; Pearson, K.J.; Marsit, C.J. Maternal Smoking during Pregnancy Is Associated with Mitochondrial DNA Methylation. Environ. Epigenetics 2016, 2, doi:10.1093/eep/dvw020.”
Reviewer Comment: Page 9, lines 321 the Authors wrote “Even though the fact that previous studies...” without references. Please add.
Response: The text now includes the following reference. “Shock, L.S.; Thakkar, P. V.; Peterson, E.J.; Moran, R.G.; Taylor, S.M. DNA Methyltransferase 1, Cytosine Methylation, and Cytosine Hydroxymethylation in Mammalian Mitochondria. Proc. Natl. Acad. Sci. U. S. A. 2011, 108, doi:10.1073/pnas.1012311108.”
Reviewer Comment: Page 10 Caption of Figure 5 is not clear. Please check.
Response: Thanks for the comments. This caption was revised.
Reviewer Comment: Page 12, lines 427-431 the Authors wrote “Notably, placental DNA methylation has been found...” without references. Please add.
Response: The text now includes the following reference. “Melzner, I.; Scott, V.; Dorsch, K.; Fischer, P.; Wabitsch, M.; Brüderlein, S.; Hasel, C.; Möller, P. Leptin Gene Expression in Human Preadipocytes Is Switched on by Maturation-Induced Demethylation of Distinct CpGs in Its Proximal Promoter. J. Biol. Chem. 2002, 277,doi:10.1074/jbc.M208511200.
Sletner, L.; Moen, A.E.F.; Yajnik, C.S.; Lekanova, N.; Sommer, C.; Birkeland, K.I.; Jenum, A.K.; Böttcher, Y. Maternal Glucose and LDL-Cholesterol Levels Are Related to Placental Leptin Gene Methylation, and, Together With Nutritional Factors, Largely Explain a Higher Methylation Level Among Ethnic South Asians. Front. Endocrinol. (Lausanne). 2021, 12, doi:10.3389/fendo.2021.809916.”
Reviewer Comment: Page 12, line 458. Reference 92 is not pertinent. The correct reference must be inserted.
Response: The text now includes the following reference. “Das, J.; Maitra, A. Maternal DNA Methylation during Pregnancy: A Review. Reprod. Sci. 2021, 28.”
Reviewer Comment: Reference 92 refers to the sentence “Prenatal maternal depression can cause covalent epigenetic changes in the DNA of their offspring that are detectable at birth in leukocytes and that may be present in other tissues, suggesting a model wherein systemic epigenetic changes may be involved in lifelong responses to the in utero psychosocial environment.” at lines 461-464.
Response: This point was fixed.
Reviewer Comment: Page 12, lines 464-466 “Maternal depression during pregnancy has been linked to an increased risk of obstetric complications such as GDM, hypertension, and PE, as well as smoking, increased alcohol consumption, and inadequate nutrition.” The reference is missing.
Response: The text now includes the following reference. “ogher, K.L.; O’Keeffe, M.M.; Khashan, A.S.; Gutierrez, H.; Kenny, L.C.; O’Keeffe, G.W. Epigenetic Regulation of the Placental HSD11B2 Barrier and Its Role as a Critical Regulator of Fetal Development. Epigenetics 2014, 9.”
Reviewer Comment: Reference 95 is not pertinent.
Response: This reference was omitted.
Reviewer Comment: Page 13, lines 482-484 “Nutrients can act directly by inhibiting...” Reference should be added.
Response: The text now includes the following reference. “Tiffon, C. The Impact of Nutrition and Environmental Epigenetics on Human Health and Disease. Int. J. Mol. Sci. 2018, 19.”
Reviewer Comment: Page 13, lines 486-487 “This, in turn, changes the expression of critical genes...” Reference should be added.
Response: The text now includes the following reference. " Choi, S.W.; Friso, S. Epigenetics: A New Bridge between Nutrition and Health. Adv. Nutr. 2010, 1.
Davis, C.D.; Ross, S.A. Dietary Components Impact Histone Modifications and Cancer Risk. Nutr. Rev. 2007, 65.”
Reviewer Comment: Page 13, lines 492-493 “Other studies...” Reference should be added.
Response: The text now includes the following reference. “Reichetzeder, C. Overweight and Obesity in Pregnancy: Their Impact on Epigenetics. Eur. J. Clin. Nutr. 2021, 75.
Jansson, N.; Rosario, F.J.; Gaccioli, F.; Lager, S.; Jones, H.N.; Roos, S.; Jansson, T.; Powell, T.L. Activation of Placental MTOR Signaling and Amino Acid Transporters in Obese Women Giving Birth to Large Babies. J. Clin. Endocrinol. Metab. 2013, 98, doi:10.1210/jc.2012-2667.”
Reviewer Comment: Page 13, line 494 “Present work...” Substitute with the name of the author “The work by...”
Response: Edited as suggested.
Reviewer Comment: Page 13, lines 498-500. Check reference 99, if appropriate.
Response: Thank you for pointing it out. The correct reference was added.
Reviewer Comment: Page 13 lines 506-509 “Low concentrations of vitamin D....” the reference is missing.
Response: The text now includes the following reference. “Claycombe, K.J.; Brissette, C.A.; Ghribi, O. Epigenetics of Inflammation, Maternal Infection, and Nutrition. J. Nutr. 2015, 145.”
Reviewer Comment: Page 13, line 512 “most defenseless” should be changed “is more susceptible”
Response: The sentence was revised.
Reviewer Comment: Page 14, line 527-528 References are missing.
Response: The text now includes the following reference. “Zenclussen, A.C. Adaptive Immune Responses during Pregnancy. Am. J. Reprod. Immunol. 2013, 69.”
Reviewer Comment: Page 14, line 535-536 References are missing.
Response: The text now includes the following reference. “Mor, G.; Cardenas, I. The Immune System in Pregnancy: A Unique Complexity. Am. J. Reprod. Immunol. 2010, 63.”
Reviewer Comment: Page 16, line 568 is reference 118 pertinent?
Response: We double-checked all references and added them as needed.
Reviewer Comment: Page 16, lines 581-584 “The role of the developmental and parental environmental exposures in shaping the ultimate metabolic characteristics of offspring has been effectively demonstrated in several human studies of populations exposed to extreme nutrient deficiencies during pregnancy. Reference should be added.
Response: The text now includes the following reference. “Shankar, K.; Pivik, R.T.; Johnson, S.L.; van Ommen, B.; Demmer, E.; Murray, R. Environmental Forces That Shape Early Development: What We Know and Still Need to Know. Curr. Dev. Nutr. 2018, 2, doi:10.3945/cdn.117.001826.”
Reviewer Comment: Page 16, lines 599-602. References are missing
Response: The text now includes the following reference. “Bleker, L.S.; De Rooij, S.R.; Painter, R.C.; Ravelli, A.C.J.; Roseboom, T.J. Cohort Profile: The Dutch Famine Birth Cohort (DFBC) -a Prospective Birth Cohort Study in the Netherlands. BMJ Open 2021, 11, doi:10.1136/bmjopen-2020-042078.”
Reviewer Comment: Page 16, lines 608-610. References are missing.
Response: The text now includes the following reference. “Schulz, L.C. The Dutch Hunger Winter and the Developmental Origins of Health and Disease. Proc. Natl. Acad. Sci. U. S. A. 2010, 107.”
Reviewer Comment: Page 17, Figure 7 “Environment factors” should become “Environmental factors”.
Response: The figure title was revised.

Reviewer 2 Report
I appreciate the authors effort in modifying the manuscript although in a very short time, but I still find inaccuracies and errors that sometimes denote the lack of knowledge in this subject but also the way to present a scientific work (figures should be revised in order to show the key message reported in the text and must be self-explanatory; tables must be carefully checked in order to report the correct information from the cited literature; the content must be revise in order to show the main findings of the cited studies). I greatly appreciate the aim of the review, fully explained and discussed in the conclusion of the manuscript. As the authors say, their article needs to focus on a review of previous studies exploring changes in maternal DNA methylation during pregnancy and associated adverse conditions, so other reported information is often unnecessary and can be removed.
Author Response
September 9, 2022
Dear Editor
We are pleased to submit our revised manuscript entitled “Epigenetics and pregnancy: conditional snapshot or rolling event” (Manuscript ID- ijms-1837533) for consideration of publication in international journal of molecular science.
We thank all reviewers for taking time to review our manuscript and for offering many constructive comments, which have contributed toward improving the quality of our paper. We have addressed all of their queries (enclose).
Sincerely
Gil Atzmon, Ph.D.
Associate Professor of Epigenetics and Genomics of Aging/Longevity
University of Haifa
Reviewer 2:
Reviewer comment: I appreciate the authors effort in modifying the manuscript although in a very short time, but I still find inaccuracies and errors that sometimes denote the lack of knowledge in this subject but also the way to present a scientific work (figures should be revised in order to show the key message reported in the text and must be self-explanatory; tables must be carefully checked in order to report the correct information from the cited literature; the content must be revise in order to show the main findings of the cited studies). I greatly appreciate the aim of the review, fully explained and discussed in the conclusion of the manuscript. As the authors say, their article needs to focus on a review of previous studies exploring changes in maternal DNA methylation during pregnancy and associated adverse conditions, so other reported information is often unnecessary and can be removed.
Response: The point raised by the reviewer is very important. We checked and revised the content in the whole manuscript including the figures and tables. Such as section 3.3 on telomerase. Additionally, we fixed all the tables and the citations that are not relevant. All the specific amendments will be addressed in the updated manuscript.
